# An X-ray micro-tomographic study of the pore space, permeability and percolation threshold of young sea ice

Sönke Maus[1], Martin Schneebeli[2], and Andreas Wiegmann[3]

[1]Department of Civil and Environmental Engineering, NTNU, Trondheim, NORWAY
[2]WSL Swiss Federal Institute for Snow and Avalanche Research, Davos, SWITZERLAND
[3]Math2Market GmbH, Kaiserslautern, GERMANY

**Correspondence:** Sönke Maus (sonke.maus@ntnu.no)

**Abstract.** The hydraulic permeability of sea ice is an important property that influences the role of sea ice in the environment in many ways. As it is difficult to measure, so far not many observations exist and the quality of deduced empirical relationships between porosity and permeability is unknown. The present work presents a study of the permeability of young sea ice based on the combination of brine extraction in a centrifuge, X-ray micro-tomographic imaging and direct numerical simulations. The approach is new for sea ice. It allows to relate the permeability and percolation properties explicitly to characteristic properties of the sea ice pore space, in particular to pore size and connectivity metrics. For the young sea ice from the present field study we obtain a brine volume of 2 to 3% as threshold for the vertical permeability (transition to impermeable sea ice). We are able to relate this transition to the necking of brine pores at a critical pore throat diameter of $\approx 0.07$ mm, being consistent with some limited pore analysis from earlier studies. Our optimal estimate of critical brine porosity is half the value of 5 % proposed in earlier work and frequently adopted in sea ice model studies and applications. By placing our results in the broader context of earlier studies, we conclude that the present threshold is more significant, in that our centrifuge experiments and high resolution 3D image analysis enable us to more accurately identify the threshold below which fluid connectivity ceases, by examining the brine inclusion microstructure on finer scales than were previously possible. We also find some evidence that the sea ice pore space should be described by *directed* rather than *isotropic* percolation. Our revised porosity threshold is valid for the permeability of young columnar sea ice dominated by primary pores. For older sea ice containing wider secondary brine channels, for granular sea ice, as well as for the full thickness bulk permeability, other thresholds may apply.

## 1  Introduction

Sea ice is a porous medium that covers, on average, 5 to 7 percent of the earth's oceans. To understand the role of sea ice in the earth system, its hydraulic permeability needs to be known. A proper understanding of the salinity evolution of sea ice requires the knowledge of its permeability (Cox and Weeks, 1988; Worster and Wettlaufer, 1997; Petrich et al., 2006; Vancoppenolle et al., 2007; Wells et al., 2013; Griewank and Notz, 2013; Turner et al., 2013; Rees Jones and Grae Worster, 2014). Through its control of the salinity of sea ice, the permeability furthermore impacts the evolution of many other physical properties like like sea ice strength and and thermal conductivity (Cox and Weeks, 1988; Worster and Wettlaufer, 1997), that depend on the brine porosity of sea ice. Of high relevance for sea ice in the climate system is also the role of permeability for the *Melt pond*

*albedo feedback*: Melt ponds from melted snow, appearing on sea ice during summer, will drain when the sea ice is permeable, exposing an ice surface that reflects more sunlight than ponded ice (e.g., Freitag and Eicken, 2003; Polashenski et al., 2017).

While permeability plays a key role for proper modelling and understanding of sea ice properties, observations are sparse and span, even at a fixed porosity, 2-3 orders of magnitude (Maksym and Jeffries, 2000). Test procedures used so far all suffer from shortcomings. Field measurements based on the filling rate of in situ boreholes only give some average measure of near-

bottom permeability. These values further depend on the unknown permeability anisotropy and pore space details, and thus are uncertain (Freitag, 1999; Freitag and Eicken, 2003; Golden et al., 2007). Laboratory studies have been restricted to relatively young and thin ice (Saito and Ono, 1978; Ono and Kasai, 1985; Saeki et al., 1986; Okada et al., 1999). To what degree these experiments resemble natural sea ice is uncertain and may only be answered by a comparison of microstructure and pore scales, not performed so far. The most frequently cited study of sea ice permeability (Freitag, 1999) was based on samples from an

ice tank experiment. Ice core segments were first centrifuged at in situ temperatures, before the permeability was obtained experimentally with a kerosene-based permeameter set-up. The advantage of centrifuging ice samples and using a liquid that does not mix with water is to avoid microstructure changes that inevitably take place during storage and/or fluid flow. However, since the study by Freitag (1999) no further observations to validate the permeability values based on this method have been published. It is thus unclear, to what degree the results are valid for natural sea ice, and how different ice growth and age might

affect the results.

The present study follows the centrifuging approach by Freitag (1999) and extends it in several ways. First, the permeability of centrifuged sea ice samples is not determined by a laboratory permeameter, but through direct numerical simulations on 3D X-ray micro-tomographic (XRT) images. Second, we perform a statistical analysis of the 3D XRT-based pore space that allows for a physical interpretation of permeability in terms of pore sizes and connectivity. Third, we extend the so far documented

porosity range to values when the ice becomes impermeable, obtain an estimate of the threshold porosity, and analyse the pore space near the threshold. Our approach allows us to revise the percolation threshold of $\phi \approx 0.05$ that has been proposed in earlier studies without consideration of the micro-structural pore size details (Petrich et al., 2006; Golden et al., 2007). Furthermore, we present a relationship between permeability and brine porosity that is valid over a wider range of porosities than so far investigated.

## 2  Field work and methods

### 2.1  Field sampling

Sea ice samples for the present study were obtained from fast ice in Adventbay of Adventfjorden, Svalbard during $14^{th}$ to $19^{th}$ April 2011, approximately 2 km from the UNIS (University Courses on Svalbard) building (Figure 1). The meteorological conditions indicate, in combination with daily ice charts from the Ice service of the Norwegian Meteorological Institute

(ftp.met.no/pub/icecharts/), that the ice was approximately 3-4 weeks old. After, most likely, freeze-up during 20.-22.04.2011, it mostly grew during a period of 10 days with temperatures around -20 °C, followed by 10 days with gradual warming. A 10 cm cover of new snow on the ice had mostly accumulated a few days prior to sampling. The insulation through the snow cover

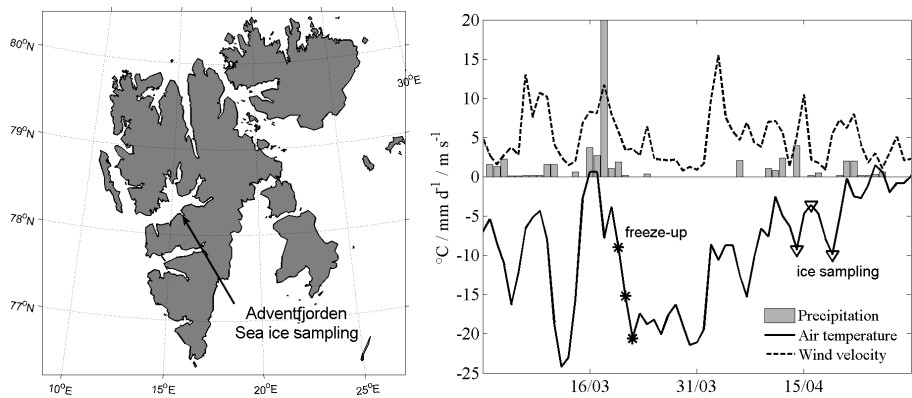

**Figure 1.** Left: Location of sampling of young sea ice in Adventbay/Adventfjorden. Right: Meteorological conditions at Longyearbyen airport in March/April 2011, from freeze-up (∗) to sampling three to four weeks later (▽)

resulted, in spite of air temperatures varying by 7 K during the sampling period, in only minor ice temperature changes over 5 days, and a temperature range of less than 1 K over 35 cm thickness. While originally sampling of ice at different temperatures was planned, the stable temperature turned out as an advantage for temperature control, and allowed to harvest and analyse ice cores of very similar salinity and structure, and rather to perform a controlled cooling sequence in the laboratory.

During each sampling date 6 full ice cores were obtained with a 7.25 cm diameter coring device (Mark III, Kovacs Enterprises) from 35 cm thick fast ice. Cores were immediately cut into 3-4 cm thick subsamples. On a first ice core, temperatures were measured with a penetration probe - this core was only used further for temperature tests. All other core segments were packed in plastic beakers, stored in an isolating box and rapidly (by snow mobiles, within 30 minutes from the beginning of coring) transported to the UNIS laboratory. For the given field conditions the temperature change that samples may have experienced during this transport might be a few tens of a Kelvin (see below in section 2.5.). We note that less isothermal ice would have required a more advanced temperature control of the different levels in the ice. At UNIS the samples were moved into temperature-controlled freezers (WAECO Coolfreeze T56) close to their *in situ* temperatures (typically within 0.3 K). During the three sampling dates a total of 15 ice cores were obtained and sectioned into 145 subsamples.

## 2.2 Laboratory cooling sequence

*In situ* sea ice temperatures were in the range -2 to -3 °C. To extend this natural range we used the following approach in the laboratory. For each of the three sampling dates one core was left at *in situ* temperatures. The sub samples from the 4 replicate cores were put into freezers controlled at lower temperatures -3, -4, -6, -8 and -10 °C and equilibrated by 1 to 3 days. The result is, for each level in the ice, a series of 5 samples with temperatures gradually ranging between in situ values (-2 to -3 °C) and minimum temperatures in the range -8 to -10 °C. In this way we generate samples with up to 4.5 times smaller brine porosity compared to the *in situ* condition.

## 2.3 Centrifugation

In a laboratory at UNIS the subsamples were centrifuged in a refrigerated centrifuge (Sigma 6K15). In our protocol the subsamples were placed on the field site into conical buckets to collect the brine that drained from them during storage. Centrifuging was performed 1 to 4 days after sampling, with longer waiting time for those samples cooled to lower temperatures. To do so, samples were placed into flexible stainless steel tea-sieves that fitted into the conical plastic buckets. Centrifuging thus extracted the brine from brine channels with a downward orientation and open to the bottom or perimeter of the sub-sample.

Centrifuging was performed at *in situ* temperatures (one core), and at the lowered temperatures from the sequence (4 cores). The centrifuged ice samples were, immediately after centrifugation set into a -80 °C freezer. On the next the mass of centrifuged brine and residual ice samples was measured. The centrifuged samples were then cut down from the initial 7.25 cm to 3.5 cm diameter, and the ice that was cut off was melted. The centrifuged brine and melted residual ice were filtered with a 100 $\mu m$ sieve, before the salinity was determined via measurements of the electrolytic conductivity and temperature (instrument WTW Cond 340i). Brine samples with salinity $> 40$ g/kg were diluted to perform the conductivity-salinity conversion with seawater standard formulas. Salinity values obtained in this way have a measurement accuracy of better than 0.2 g/kg.

A duration of 15 minutes and a centrifuge acceleration of $40 \times g$ (earth gravity) was selected for centrifuging. These numbers have been chosen due to several aspects of the approach. The acceleration ensures a pressure force ($40\rho_i gH$) of less than 15 $kPa$, well below the lowest tensile strength values (20-50 kPa) observed for natural sea ice (Weeks, 2010). This ensures that samples do not deform internally during centrifuging, though it could not prevent the compression and micro-fracture of the fragile ice-seawater interface sub-sample. Second, it is important to set the centrifuge temperature close to but slightly (0.5-1 K) below the sample temperature, because otherwise the samples may warm up in the end and release additional brine. A third aspect, the impact of parameter choice on proper brine removal, is discussed below in connection with the permeability simulations.

As discussed in earlier applications (Weissenberger et al., 1992; Freitag, 1999; Krembs et al., 2001) centrifuging gives important information about the disconnected and connected fractions of the brine pore space. The centrifuged porosity may be associated with the *effective porosity* $\phi_{eff}$ relevant for fluid flow/permeability (Freitag, 1999), for which we shall use $\phi_{en}$ henceforth. Let the total brine porosity $\phi$ be the sum of the centrifuged brine porosity $\phi_{cen}$ and the residual brine porosity $\phi_{res}$. Assuming that the corresponding brine salinities are the same, these porosities may be determined from salinity determinations alone:

$$\phi_{cen} = 1 - \phi \frac{S_{ir}}{S_i} \left( \frac{S_b - S_i}{S_b - S_{ir}} \right), \quad \phi_{res} = \phi - \phi_{cen} \tag{1}$$

where $S_i$ is the bulk salinity of the original ice sample, $S_{ir}$ the residual salinity of a sample after centrifugation and $S_b$ the salinity of the centrifuged brine, and $\phi$ was determined from $S_i$ and temperature $T_i$, assuming thermodynamic equilibrium and applying equations from Cox and Weeks (1983).[1] The centrifuged and cut ice samples were further stored in a -80 °C low

---

[1] Alternatively $\phi_{cen}$ and $\phi_{res}$ can be computed from the mass and salinity of the centrifuged brine and ice samples (Weissenberger et al., 1992). We tested also this approach and only found relative differences of a few percent.

temperature freezer, and and kept for 2 months at this temperature (including transport on dry ice). 2 days prior to imaging by X-ray micro-computed tomography (XRT) described below the samples were equilibrated to -20 °C.

The centrifuge parameters depend on centrifuge type and were carefully chosen on the basis of several tests. (i) Ice samples were centrifuged with temperature loggers to determine temperature stability. Slight warming of the centrifuge was observed, leading us to the choice of an in initial centrifuge temperature 1K below the ice in-situ temperature. A similar value was chosen by Weissenberger et al. (1992) for similar centrifuge times. (ii) Varying the centrifuging time from 10 to 20 minutes showed that more than 95% of brine where extracted during the first ten minutes, and we selected 15 minutes. (iii) Freitag (1999) noted that incomplete centrifugation of brine might lead to brine remnants which, after cooling and freezing, might block pores and decrease the permeability. We have indeed found such a result in an earlier study with centrifuge acceleration of 15 g (Buettner, 2011) and thus tested the effect of relative centrifuge acceleration for three ice cores at 10, 25 and 40 g. The result was on average 20% less centrifuged brine at 10 g, but only a slight non-significant 5% difference between 25 and 40 g. We thus are confident that 40 g is a proper choice for extracting the connected brine.

The centrifuged brine mass on which the effective porosity is based also includes brine that has leaked from the sample during storage, prior to centrifuging. In our study this pre-drained brine volume was not negligible and contributed on average 28% of the total (leaked and centrifuged) brine volume. On the one hand this value may be an overestimate, as it could include small ice particles that fell into the cup during sampling. On the other hand, there is very likely some brine lost during coring and cutting, which will underestimate the centrifuge-based effective porosity. Both effects imply a difference between CT-based and centrifuge based estimated of effective porosity that we cannot resolve with our data.

## 2.4   X-ray micro-tomography

X-ray tomographic imaging was performed at the WSL Swiss Federal Institute for Snow and Avalanche Research, Davos, Switzerland, with two desktop cone-beam microCT instruments (MicroCT 40 and MicroCT 80, Scanco Medical AG) that operate with a microfocus X-ray source (7 $\mu$m diameter) and detectors of 2048 x 256 and 2048 x 128 elements, respectively. The instrument was located in a cold room at -20 °C. However, the temperature within the CT chamber was slightly higher, -16 °C. The samples, after centrifugation reduced to 35 mm diameter, were again slightly reduced to fit into the 35 mm diameter sample holders, and then scanned with a 37 mm field of view, yielding a nominal pixel size of $\approx 37000/2048 = 18\mu m$. Scanning time was roughly 1 hour per centimetre sample height, and thus 3-4 hours per sub sample. 1000 images per 360 degrees rotation were obtained. For the image analysis. horizontally 1200 x 1200 pixels were cropped from the center of the 2048 x 2048 field of view, and 1500 pixels vertically. The resulting XRT grayscale images were stored as 16-bit stacks, and filtered with ImageJ (rsb.info.nih.gov/ij/), applying a 2 pixel median and Gaussian blur filter (standard deviation 1.5).

The image segmentation into air, ice and brine was also performed with ImageJ as illustrated in Figure 2 for two samples, one with high air and low residual brine porosity and another one with low air and moderate brine porosity. Our approach to find the air-ice and ice-brine thresholds was as follows. First, brine was ignored, and a global threshold that separates air and ice was found on sub-images with approximately equal fractions of air pores and ice, using Otsu's method (Otsu, 1979). Comparison with manual segmentation of single thresholds indicates an accuracy of 0.5 to 1% for the air porosity, being higher

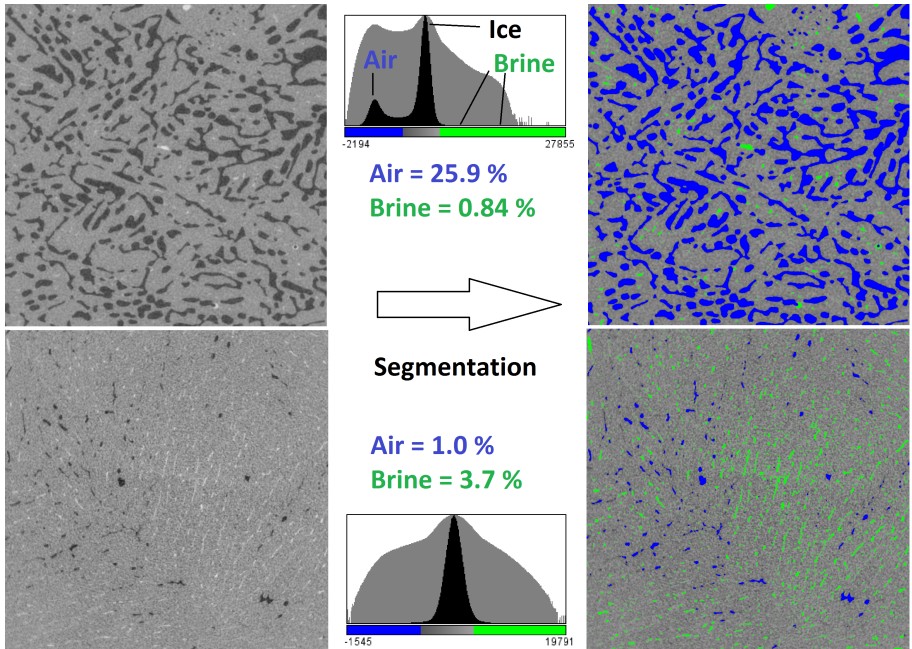

**Figure 2.** Segmentation of pre-filtered grey scale absorption images (left) into the classes of air (blue), brine (green) and ice (grey) for a high porosity (upper) and low porosity (lower) image. The images in the middle give the histograms of grey values with a linear (black) and a log (grey) scale. Upper example: a high porosity image with a well-defined air peak, yet little entrapped brine. Lower example: a sample with more entrapped brine than air/centrifuged brine.

at high porosities. For the ice-brine threshold it was more difficult to find an automated procedure. Segmentation with Otsu's algorithm gave generally too high brine content (compared to direct measurements). The best results, comparing to measured salinity, were obtained within ImageJ using the so called Triangle algorithm, yet still with considerable scatter. It was thus decided to rather use an empirical approach that sets the ice-brine threshold to 1.20 times the grey value of the ice mode. This number led to least deviation of average CT-derived salinity and the salinity of melted samples. However, also here the absolute uncertainty of brine volume was 0.5 to 1%, corresponding to a relative uncertainty of $30 - 100\%$, as residual brine porosities at -16 °C were low. Hence, the relative accuracy in residual brine determination is much smaller than for the air porosity. As most likely explanation it is considered that brine inclusions were not much larger than the voxel size, and that brine is often found together with tiny air bubbles. This implies a considerable number of mixed air-brine pixels that have a grey value just a bit larger than ice.

Hence, for the present pore sizes, spatial resolution and brine salinity ($\approx 185$ g/kg at -16°C) there are considerable uncertainties in ice-brine segmentation, and an unsupervised approach (i.e. without setting a threshold based on alternative bulk salinity measurements) was not found. However, the permeability is little or not affected by the residual brine, and rather relates to the open air porosity (centrifuged, connected brine) that was determined with reasonable accuracy.

The imaged samples, cylinders with 35 mm diameter and 25-30 mm height, were again subdivided vertically into samples of 5.5 mm height. The subdivision is important for proper determination of permeability and percolation, as for 20 mm high samples a considerable fraction (10-30%) of slightly (10 to 30 degrees) inclined pores are running off at the vertical sides. Examples of these 3D sub-images to be analysed here are shown in the sections below. With the current imaging settings, and image processing for analysis and simulations, we expect to observe pores and inclusions with smallest dimensions of 36 $\mu$m (corresponding to a Nyquist criterion of 2 times the voxel size of 18 $\mu$m). This is an improvement by a factor of two compared to the voxel size of $41.5\mu$m in the CT-image study of laboratory grown ice by Pringle et al. (2009). Our horizontal scale is large enough to also observe pores and brine channels situated between grains of typical dimensions 5 to 20 mm. Our standard vertical scale of 5.5 mm is smaller than used in standard sea ice bulk sample analysis of several centimetres, yet we can always merge the subsamples to look at comparable vertical scales. The choosen horizontal and vertical sample scales are well above the typical pore scale characteristics of young ice obtained by Eicken et al. (2000) based on the analysis of Magnetic Resonance Images with lower resolution (0.09mm voxel size).

## 2.5 Sampling, transport, storage and textural changes

Special care was taken to minimise undesired temperature changes and variability prior to centrifugation and imaging. The cut samples of the relatively isothermal sea ice were transported in an isopor box (inside a larger insulated aluminium box) to the laboratory. Transport and sorting into small temperature controlled freezers happened within half an hour. As each sub sample was packed in a conical plastic cup, temperature changes are, due to the large effective specific heat capacity, considered negligible. The box temperature was logged by a temperature logger, as well as temperatures were directly measured on samples, being within 0.2 K of in situ values. The next step, cooling of sub samples in the laboratory, took place within these freezers set to lower than in situ temperatures. With samples within the plastic cups cooling rates (with most heat loss due to internal freezing) were moderate and in the range 1-5 K per day, comparable to natural cooling rates. An important aspect of the approach was also that samples were only cooled, not warmed. This avoids the known hysteresis, that brine expelled during cooling is not reintroduced into a sample upon warming.

Though we have no strict proof for this, we believe that microstructure changes during 1 to 2 days of close to isothermal storage are minor (this is based on unpublished work of repeated scanning). More relevant could be effects due to freezing and redistribution of brine. First, one could expect that simultaneous cooling of sub samples from all sides may redistribute brine in a way that differs from mostly vertical heat loss of ice in the field. We do not find brine accumulation in the center of samples, indicating that also the multi-directional sample cooling redistributes brine along the predominantly vertically oriented pores. Brine could be redistributed vertically in some non-uniform way within a 3 cm thick sub sample, and implications will be considered in the discussion. Second, we treat our sample isothermally, which is justified as the in situ temperature profile suggests a difference of 0.1 K along the vertical direction. Third, sample storage after centrifugation at low temperature (-80 °C) has likely led to almost complete precipitation of all residual brine. During XRT imaging these salt crystals have dissolved again. As the microstructure of these pores will very likely differ from field values, we do not analyse it here. We regard it as unlikely, that this hysteresis of disconnected pores has affected the networks of connected pores.

We finally note that the small in situ temperature range made this study logistically easier as if the ice had been sampled during a cold period.

## 2.6 Permeability simulations

Flow through porous media at relatively low velocities is governed by Darcy's equation (Dullien, 1991; Nield and Bejan, 1999). In one dimension

$$\overline{V} = \frac{K}{\mu}\frac{dP}{dz}, \tag{2}$$

gives the dependence of average velocity $\overline{V}$ (discharge per unit area) on pressure gradient $dP/dz$, dynamic viscosity $\mu$ of the fluid and permeability $K$. The latter has dimensions of area and may be imagined as the cross section of an equivalent channel of fluid flow through the pore space. The present approach to obtain $K$ is to centrifuge the brine from the pore space, store the samples, and later perform permeability experiments (Freitag, 1999) or CFD simulations (Maus et al., 2013). It is thus of interest how the settings during centrifuging may impact the results: Consider the pressure gradient $dP/dz = \rho_b g$ across a sample filled with brine of density $\rho_b$. Inserting this into Equation. 2 one obtains the relationship

$$\overline{V} = K\frac{g}{\nu}, \tag{3}$$

where the kinematic viscosity $\nu$ has replaced $\mu/\rho_b$. This equation actually states the conversion from hydraulic permeability $K$ to hydraulic conductivity $\overline{V}$. Replacing this bulk flow $\overline{V}$ by $\phi_{eff}V$, where $V$ is the actual velocity within pores contributing to the flow, the condition for brine removal from a sample of height $H$ during time $t$ requires $\overline{V} > \phi_{eff}H/t$. Further replacing $g$ by the effective $g_{eff}$ in the centrifuge, we can write Equation. 3 as

$$K > \phi_{eff}\frac{H}{t}\frac{\nu}{g_{eff}}, \tag{4}$$

as condition for full removal of brine during centrifuging. With $\phi_{eff} = \phi = 0.024$ at the lower end of our porosity range (see below), sample thickness H=0.04 m, centrifugal time t = 9000 s, $\nu = 3.2 \times 10^{-4}$ m$^2$s$^{-1}$ (at -10 °C) into Equation. 3, and $g_{eff} = 40g$, we obtain $K > 9 \times 10^{-15}m^2$. In ice samples with a lower permeability than that value, one can expect incomplete removal of brine. Upon cooling this brine will partially freeze and may render the sample impermeable. As we will find below, the lowest simulated permeability value in our study is $8.1 \times 10^{-15}m^2$, close to the latter estimate. However, below we also find that $\phi_{eff}$ is much lower than $\phi$ when low porosities are approached, decreasing this limit for $K$ by at least a factor of 3-4. We thus assume that insufficient brine removal during centrifuge acceleration is not not a large problem for our results.

Here we report on vertical permeability computations that have been performed with GeoDict "Geometric Material Models and Computational PreDictions of Material Properties" GeoDict (2012-2020). The simulations were run on the mentioned sub-images of 1200 x 1200 x 300 voxels (300 corresponds to 5.5 mm height), with the SimpleFFT solver of the FlowDict module in GeoDict. The solver obtains, for a given pressure drop across the sample, the stationary fluid flow on a uniform grid based on the iterative solution of the Stokes-Brinkman equation (Wiegmann, 2007; Cheng et al., 2013; Linden et al., 2018). Recent work has demonstrated the quality of the numerical solution of the FlowDict solver in comparison to observations (Zermatten

et al., 2011; Gervais et al., 2015; Gelb et al., 2019). In our setup a 10 pixels thick inflow region at the top and bottom of the sample was used in connection with periodic boundary conditions. Computations on 3D images of dimension 300 x 1200 x 1200 ($\approx$ 5.5 x 22 x 22 mm) required typically 25-30 GB RAM. Limiting the accuracy to 1% appeared to be sufficient for most samples to converge in between 200 to 600 iterations, which took typically 1 to 3 days per sample on a 4 core pc with a 3 GHz CPU. Simulations performed for 150 images have been published (Maus et al., 2013). The latter results have been revised in the present study and simulations have been repeated for those samples that had not reached the convergence criterium (mostly low porosity/permeability samples). With currently faster hardware, and improvement of the Geodict solver, simulations are nowadays 10 to 20 times faster.

## 2.7 Pore space analysis

The permeability $K$ of a porous medium (equation 2) is often parametrised in terms of total porosity in the form $K \sim \phi^b$, where the range $2 < b < 5$ has been found in observations (Dullien, 1991; Happel and Brenner, 1986). A more concise and physically consistent formula for the permeability is (Paterson, 1983, e.g.,)

$$K = a\tau^2 D_c^2 \phi_{eff}^b \tag{5}$$

wherein $\phi_{eff}$ is the effective porosity for fluid flow, $D_c$ a characteristic pore diameter, $\tau$ tortuosity of the flow path and $a$ a constant. For simple flow geometries this relationship is exactly known. E.g., for a bundle of parallel cylindrical pores of diameter $D_c$ with cross sectional area $\phi = \phi_{eff}$ one has $\tau = 1$ and $a = 1/32$, while a system of parallel vertical lammellae (flow through slits of width $D$) with $\phi = \phi_{eff}$ one has $a = 1/12$ (Paterson, 1983; Dullien, 1991, e.g.,). In more complex networks with a distribution of pore sizes one has to find a characteristic pore scale $D$, tortuosity $\tau$ and $a$ will depend on the detailed network morphology. Here we shall investigating how $D$, $\tau$ and $\phi_{eff}$ all depend on total porosity, to understand for which regime a simplified relationship $K \sim \phi^b$ is valid.

### 2.7.1 Porosity and volume fractions

In the present study we shall neglect solid salts. Including solid salts in the calculations would decrease brine volume fractions at the lower end of our porosity range by 0.1-0.2 % (Cox and Weeks, 1983, see), and have little effect on our results. We divide the total porosity of sea ice into the volume fractions of brine $\phi$ and air $\phi_a$. The porosity metrics relevant for our study are summarised in Table 1. In general, the brine porosity $\phi$ is considered as the sum of a connected (infinite cluster) part $\phi_{eff}$ and a closed (disconnected) part $\phi - \phi_{eff}$. We use both $\phi_{cen}$ from centrifugation and $\phi_{opn}$ from the CT image analysis to estimate the open porosity. The closed brine porosity can be obtained either from the salinity of melted centrifuged samples $\phi_{res}$ or by image analysis $\phi_{cls}$. Air porosity $\phi_a$ is only available through CT image analysis.

The open and closed fractions may be scale dependent. E.g., a closed cluster of brine inclusions could appear open in small samples. Another effect of finite sample sizes is that, because channels are not strictly vertical, some are running out to the sides. The porosity metric relevant for the permeability simulations is thus the volume fraction that connects the upper and lower side of an ice sample, henceforth noted as connected porosity $\phi_{zz}$.

**Table 1.** Porosity metrics and their determination

| Porosity metric | symbol | Centrifuging | XRT imaging |
|---|---|---|---|
| Total brine porosity | $\phi$ | Sum of $\phi_{cen}$ and $\phi_{res}$ | Sum of $\phi_{opn}$ and $\phi_{cls}$ |
| Open brine porosity (infinite cluster) | $\phi_{eff}$ | $\phi_{cen}$ – centrifuged brine volume fraction | $\phi_{opn}$ – Imaged air volume fraction (centrifuged brine) open to any sample side |
| Closed brine porosity | $1 - \phi_{eff}$ | $\phi_{res}$ – residual brine volume fraction based on salinity $S_{ir}$ and temperature $T$ | $\phi_{cls}$ – Imaged volume fraction of closed brine pores (converted to in situ temperature) |
| Connected brine porosity (here: vertically) | $\phi_{zz}$ | not determined | $\phi_{zz}$ – Imaged air volume fraction (centrifuged brine) open to both vertical sides |
| Air porosity (closed) | $\phi_a$ | not determined | $\phi_a$ – Imaged closed air volume |

For the air porosity we assume that all air is contained in closed air inclusions entrapped in the ice and just define one $\phi_a$ term. Air bubbles contained in open brine pores are not detected by our approach, as they likely are centrifuged out with the brine.

We obtain the porosity metrics $\phi_{opn}$, $\phi_a$, $\phi_{zz}$ and $\phi_{cls}$ with the GeoDict module Porodict, that can be set to determine for any material the porosity open to a specific side (we use all sides) of a CT image. While $\phi_{opn}$ and $\phi_{zz}$ are associated with the centrifugation temperature (-2 to -10 °C), the porosity fraction $\phi_{cls}$ has to obtained from the brine porosity $\phi_{cls16}$ imaged at the CT operation temperature (-16 °C) by using the equations from Cox and Weeks (1983).

### 2.7.2 Pore size characteristics

We define characteristic pore scales in correspondence to the different porosity metrics, given in table 2. For the pore space analysis we also used the GeoDict module Porodict. It offers two algorithms to obtain a pore size distribution (GeoDict, 2012-2020).

The first uses a sphere fitting algorithm to determine the fraction of the pore space that belongs to a certain diameter interval. The algorithm thus determines the minor axis of a cylinder with elliptical cross section. This is done for open and closed air and brine pore classes. From the distribution we obtain the median, in terms of volume. The results from this analysis are the *open brine pore size* $D_{opn}$, the *closed brine pore size* $D_{cls}$ and the *air pore size* $D_{air}$.

The second algorithm is based on the virtual injection of spheres into the sample to determine the fraction of the pore space that can be accessed through a sphere of a given radius. The latter is a *porosimetry* (Dullien, 1991, e.g.,) algorithm that determines an effective volume distribution of pores limited by throats, and is termed *throat size* in the following. We obtain

**Table 2.** Characteristic pore scales

| Length scale [a] | symbol | CT image analysis approach |
|---|---|---|
| Open brine pore size | $D_{opn}$ | Median of the open air pore size distribution |
| Closed brine pore size | $D_{cls}$ | Median of the closed brine pore size distribution |
| Air pore size | $D_{air}$ | Median of the closed air pore size distribution |
| Brine pore throat size | $D_{thr}$ | Median of open air throat size distribution, determined by virtual porosimetry |
| SSA length scale | $D_{ssa}$ | Specific surface area (SSA) length scale, obtained from SSA assuming all pores are infinite circular cylinders: $D_{ssa} = 4SSA/\phi$ |
| Maximum path diameter | $D_{pth}$ | Diameter of the path that connects the sample surface and bottom (vertical direction) |
| Maximum path length | $L_{pth}$ | Length of the path that connects the sample surface and bottom (vertical direction) |

[a] All median values are volume-based

the *median throat size* $D_{thr}$ as a global measure, by injection of spheres from all 6 directions neck or throat sizes in the pore network.

We obtain two further characteristic length scales. One is the maximum path diameter $D_{pth}$, which is the maximum diameter of a sphere that can pass through the sample. This lengthscale is of interest for the permeability, and it is easy to determine. The second lengthscale is based on the specific surface area (SSA) of the samples (here defined as internal surface per sample volume), also determined in PoroDict. If all pores are uniform and parallel cylinders, then their diameter may be related to $SSA$ through $4\phi/SSA$, which shall be defined here as length scale $D_{ssa}$.

## 3 Results

### 3.1 Temperature and salinity

The ice thickness (35 cm on average) did not change measurably during our sampling period, the thickness range was 33 to 36 cm for the 18 cores obtained. Here we focus mostly on the cores obtained on 16.04. and all cores from this date were CT-scanned and analysed using the methods described above. The ice had a surface (ice-snow interface) temperature of -2.9 °C and a near bottom interface near the freezing point of seawater (-1.9 °C). Note that, due to a 10 cm snow cover, the ice temperature from the other two sampling dates, two days earlier and later, was very similar. Figure 3a shows the *in situ* temperature profile as well as the temperatures to which the 5 microstructure cores were lowered prior to centrifugation. Figure 3b and c show the corresponding salinity profiles and brine volume profiles. For the brine volume also the *in situ* values are given as black dots.

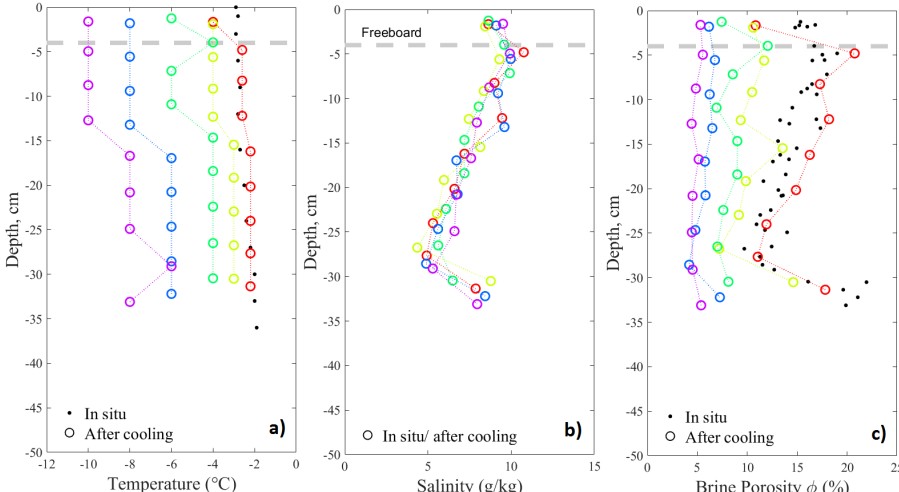

**Figure 3.** Properties of 5 + 1 sea ice cores obtained on 16.03.2011 in Adventbay, Svalbard. a) In situ ice temperature shown as black dots, 5 lowered centrifugation temperatures as coloured circles. b) Bulk salinity obtained from centrifuged brine and melted residual ice in colors. c) Brine porosity $\phi$ based on thermodynamic equilibrium shown. Black dots show in situ values, while coloured circles give the brine porosity for the lowered centrifugation temperatures.

The salinity of the ice was obtained from mass and salt balance of the centrifuged brine and the cut residual ice samples. We also obtained salinity profiles for the earlier (14.03.) and later (19.03.)sampling dates (and did the same cooling and centrifuging experiments). As for the ice thickness the salinity did not change measurably during this period. The salinity profile shows the well known C-shape, with some indication of drainage at the very surface above the freeboard [2]. All 5 salinity profiles are very similar and show little variability. This gives confidence that the temperature dependence during our cooling sequence, not internal variability of the cores, will dominate the results.

## 3.2 Porosity

### 3.2.1 Centrifuged porosity

In Figure 4a the centrifuged brine porosity $\phi_{cen}$ obtained in the centrifuge experiment is shown in dependence on the total brine porosity $\phi$. This plot is based on all 15 ice cores from the 3 sampling dates, and thus 145 subsamples of 3-4 cm thickness. The data indicate that at a certain total porosity $\phi_c$ the $\phi_{cen}$ becomes zero. To find this threshold we have regressed $\phi_{cen}$ against $(\phi - \phi_c)^{\beta}$ to obtain the optimum pair of $\phi_c$ and the critical exponent $\beta$. The result is the equation

$$\phi_{cen} = 0.569(\phi - 0.024)^{0.832} \tag{6}$$

[2]Note that, what is indicated as 'freeboard' in the figure, refers to the ice without the freshly fallen snow

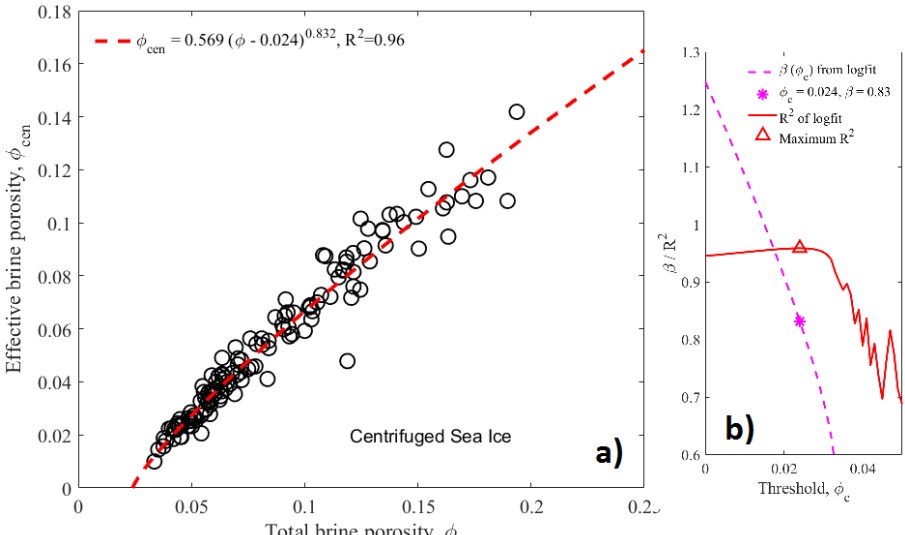

**Figure 4.** a) Relationship between centrifuged brine porosity $\phi_{cen}$ and total brine porosity $\phi$, based on 3-4 cm thick samples from 15 young ice cores of 35 cm length. b) Optimum exponent $beta$ in dependence on porosity threshold $\phi_c$ and the $R^2$ of double-logarithmic least square fits of $\phi_{cen}$ versus $(\phi - \phi_c)^\beta$. The point of maximum correlation is shown as a star.

and shown in Figure 4a as a red dashed curve. Figure 4b shows the regression results in terms of the dependence of $\beta$ on $\phi_c$. The red curve shows the maximum $R^2$ corresponding to this $\beta(\phi_c)$ curve. The maximum $R^2 = 0.96$ is found at $\phi_c = 0.0240$. For the exponent $\beta = 0.832$ the 95% confidence bounds from the log-fit are [0.803, 0.861]. For the critical $\phi_c$ we obtain confidence bounds by using $0.803 < \beta < 0.861$ and $\phi_c = 0.0240$ as input to a power law regression, which in turn resulted in 95% bound range of $0.20 < \phi_c < 0.29$. Note that a linear fit with $\beta = 1$ would give $\phi_c = 0.011$, as we calculated earlier (Maus et al., 2013). The present analysis shows that the critical exponent $\beta$ differs significantly from one.

### 3.2.2 CT-based open porosity

The CT imagery allows us to view the morphology of closed and open pores in some detail, which is illustrated in Figure 5a to c. For better visibility 5a is cropped from the center of the original image (to 1/2 horizontally). Ice is made invisible to illustrate the disconnected (in green) and connected pores (in red). Connected is here used synonymously to open, that is the pore is open to any of the 6 lateral boundaries of the 3-d image.

The horizontal slices are taken from two different regimes of this image, one with predominately connected (5b) and one with a similar fraction of connected and disconnected pores (5c). In the predominately connected Figure 5b one observes a high degree in horizontal connectivity. The patterns appear well resolved by the present voxel size (of $18\mu m$). One also can see that there are many bottlenecks (or throats) in the horizontal connectivity, and one can identify some green spots, where inclusion shave pinched off. In Figure 5c with many more disconnected inclusions, the overall connected pore width is smaller and the horizontal connectivity is low. Note however, that the red pores are still connected to one of the sides of Figure 5a.

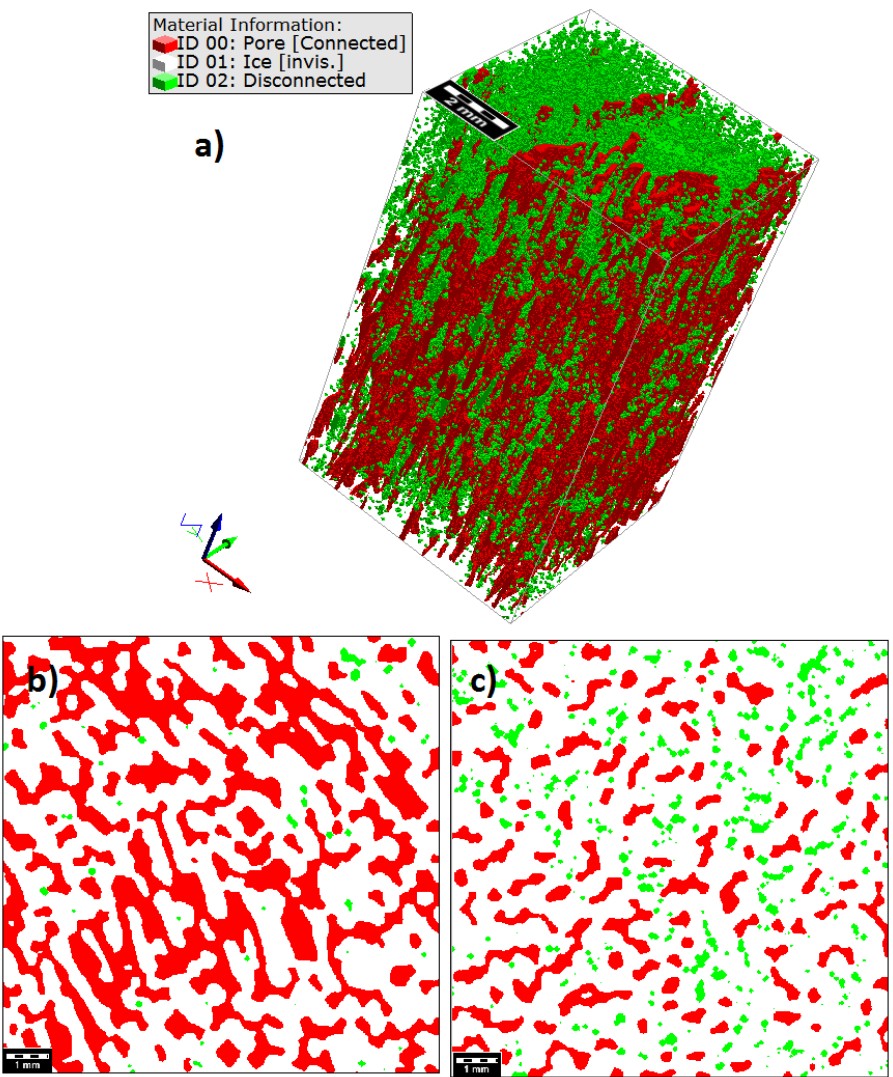

**Figure 5.** XRT micro-tomographic images illustrating the open (red) pores and closed (green) brine pores in young sea ice. a) 3D image, ice being invisible; b) horizontal section from the 3D image with open pores dominating, ice being white; c) horizontal section from the 3D image with a similar fraction of open and closed pores. The 2D scale bar is 1 mm, the side length 10.8 mm.

In Figure 6a we have plotted the CT-based open brine porosity $\phi_{opn}$ against the CT-based total brine porosity $(\phi_{opn} + \phi_{cls})$ for all samples, and compare them to the centrifuge relationship between $\phi_{cen}$ and $\phi$. Figure 6b shows the corresponding centrifuge data for the same 5 ice cores and sampling day, also on a double logarithmic scale (note that Figure 4 and the optimal fit was based on all 15 ice cores from the three sampling dates). It is obvious that the smaller CT samples extend the range from the centrifuge data to lower $\phi$ and $\phi_{opn}$ compared to $\phi_{cen}$.

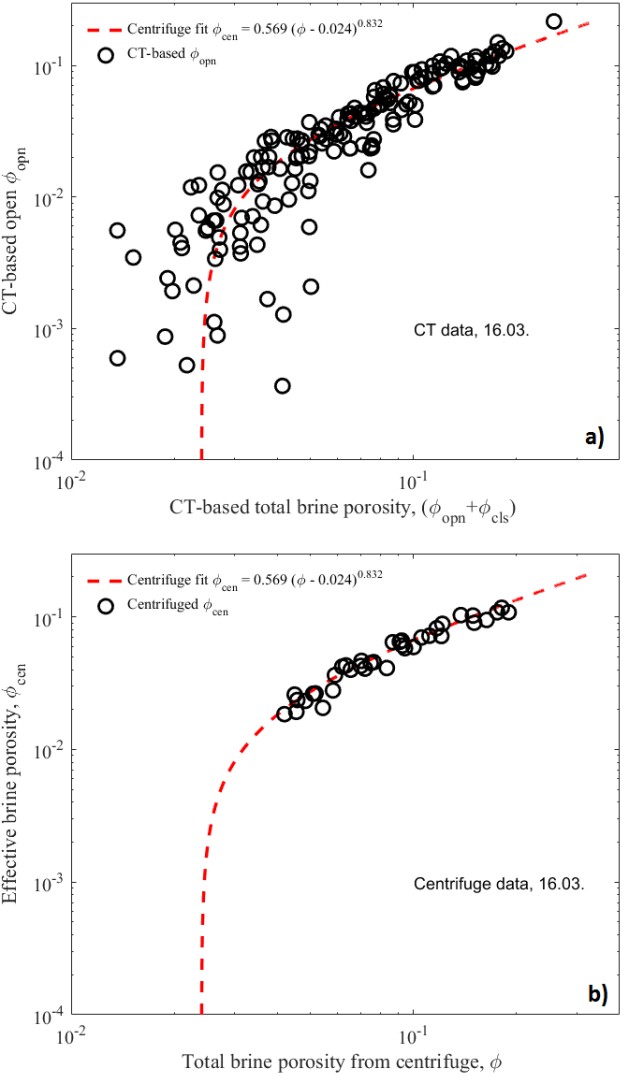

**Figure 6.** a) Relationship between CT-based open brine porosity $\phi_{opn}$ and CT-based total brine porosity $(\phi_{opn} + \phi_{cls})$, shown as circles, in comparison to the centrifuge based effective porosity fit (dashed red curves. b) the relationship between $\phi_{cen}$ and $\phi$ for the same ice core samples (a 5 core subset of the 15 ice cores in Figure 4a). Note that a) shows results for for sub-samples of the samples in b).

Above a total porosity of $\phi \approx 0.05$ the CT-based open porosity agrees well with the $\phi_{eff}(\phi)$ relationship obtained by centrifugation. At lower porosities the CT data still follow the relationship reasonably, in support of the deduced threshold

$\phi_c = 0.024$, yet become more scattered. Note that each CT data point represents for a 2 x 2 x 0.55 $cm^3$ sub-sample, roughly 1/50 of the volume of the centrifuged samples. The scatter may thus be related to centimetre scale internal variability. On the other hand, the scatter may be due to segmentation errors for both $\phi_{opn}$ and $\phi_{cls}$ that at low porosities may reach hundred percent.

### 3.2.3   Centrifuge-based open porosity conversion

The CT image based permeability and pore sizes to be presented in the following paragraphs could be correlated to different porosity metrics - the centrifuge effective porosity $\phi_{cen}$, the centrifuge-based total porosity $\phi$, the CT-based effective and open $\phi_{opn}$ and the CT-based total brine porosity $(\phi_{opn} + \phi_{cls})$. To make a comparison to other studies, and a general application feasible, the total brine porosity is chosen. However, permeability will depend on the CT-based open porosity $\phi_{opn}$ or more accurately the connected porosity $\phi_{zz}$. The CT porosities in Figure 6a are scattered, and we cannot say to what degree this
is due to segmentation errors, small undetected small brine inclusions, and/or redistribution during cooling of the centrifuged sample, in particular for $\phi_{cls}$. We thus make the following approach. We use the open porosity $\phi_{opn}$ in connection with the centrifuge best fitted equation 6, to obtain a CT-based total brine porosity. Hence, all data in Figure 6a are mapped onto the red dashed curve. In this way we preserve the essential property information to which the permeability relates, the open porosity $\phi_{opn}$, but present all data in terms of a total brine porosity $\phi$ that is computed from $\phi_{opn}$.

## 3.3   Permeability

The results of the permeability simulations for all sub samples are shown in Figure 7 in relation to the total brine porosity $\phi$ (converted from $\phi_{opn}$). As noted, this conversion incorporates the percolation threshold $\phi_c = 0.024$ deduced from the centrifugation, into the analysis. The simulations span a permeability range from $8 \times 10^{-15}$ to $7 \times 10^{-9}$ m$^2$. The porosity regime 0.024 to 0.33 above the percolation threshold is shown with grey shading. In this regime we found also impermeable samples.

We obtain two relationships between permeability $K$ and total brine porosity $\phi$ by double-logarithmic least square fitting. Due to the extreme values we use the robustfit.m Matlab function that gives less weight to outliers. Also, only the data with $\phi > 0.031$ was fitted, that is the regime where no impermeable samples are found. The first relation is a simple power law as most frequently used in sea ice studies involving the permeability:

$$K = 1.7 \times 10^{-7} \phi^{4.0} \ m^2 \tag{7}$$

The 95 % significance bounds are a factor of $10^{0.5}$ for the pre-factor and 0.4 for the exponent (Figure 7). The second is a percolation-based relationship between between $K$ and $(\phi - \phi_c)$

$$K = 1.7 \times 10^{-8} (\phi - 0.024)^{2.6} \ m^2 \tag{8}$$

Also here, only the data with $\phi > 0.031$ was fitted, trough the relationship is shown for the whole regime to illustrate the percolation behaviour. The 95 % significance bounds are here a factor of $10^{0.4}$ for the pre-factor and 0.3 for the exponent. The
355 $R^2$ of both fits is almost the same. However, Equation 7 does not account for the transition to impermeable samples at low porosities.

At a given porosity the permeability can typically vary over two orders of magnitude. There are, however, a couple of data points with larger deviation. Figure 8 illustrates the different microstructures to which this behaviour is related. Three examples of sample types have been selected: Type (I) is the most frequent sample type of young ice, with many parallel

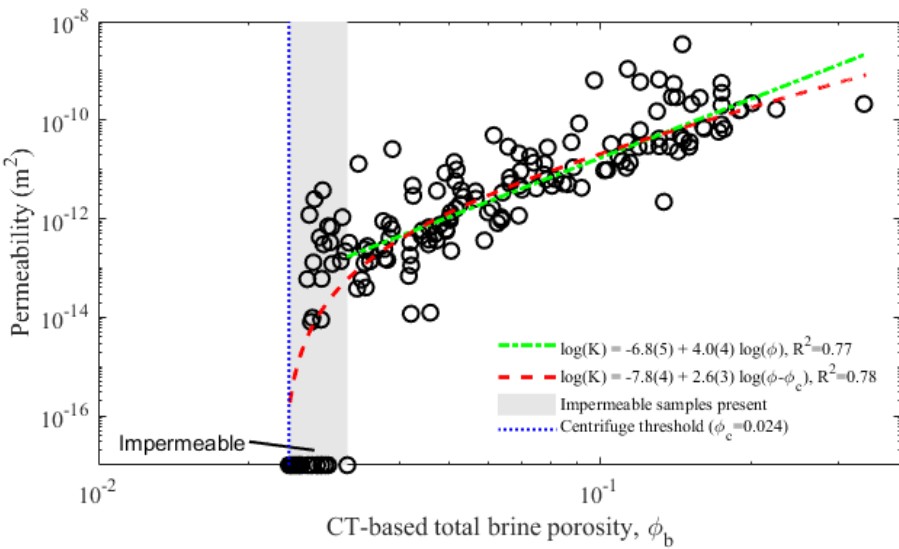

**Figure 7.** Relationship of simulated vertical permeability $K$ and the total brine porosity $\phi$. Two log-log fits are drawn and specified in the legend, corresponding to power laws of the form $K = a(\phi)^b$ (green curve), and $K = a(\phi - \phi_c)^b$ as red curve. The numbers in brackets are the uncertainties in the last decimal of the log-log least square fit. The grey shading indicates the regime where permeable and impermeable samples are found.

vertically oriented layers of pores and inclusions. The vertically connected pores are distributed over the whole sample (with total $\phi \approx 10\%$). The computed permeability ($K = 1.0 \times 10^{-11} m^2$) is close to the least squares fits. Type (II) is a sample type, with a rather localised concentration of vertically connected parallel layers and pores. The example has a total brine porosity $\phi \approx 3.3\%$) slightly above the percolation threshold, and the computed permeability ($K = 1.4 \times 10^{-11} m^2$) is 2 orders of magnitude above the fitted relations. Type (III) is a sample type with very low brine fraction of connected pores ($\phi_{zz} \approx 0.06\%$). The example has a total brine porosity $\phi \approx 3.4\%$ slightly above the percolation threshold, and the computed permeability ($K = 4.2 \times 10^{-14} m^2$) closely follows the least squares fit.

### 3.3.1 Connected porosity and tortuosity

According to equation 5 the permeability simulations are related to two additional properties. The first is the connected porosity $\phi_{zz}$, the second the tortuosity $\tau$ of the flow across the sample. For finite size images these may be related in the following way: If the tortuosity approaches or becomes larger than the sample size, then $\phi_{zz}$ will decrease, because channels will hit the lateral boundaries. Both properties are therefore investigated in Figure 9a and b.

Also for the connected porosity in Figure 9a we obtain a double-logarithmic fit of the form $\phi_{zz} \sim (\phi - \phi_c)^b$ and find an exponent $b = 1.2 \pm 0.1$ that is larger than the exponent $0.83 \pm 0.03$. Comparing the fits in the figure shows that $\phi_{zz}$ is consistently bounded from above by $\phi_{cen}$.

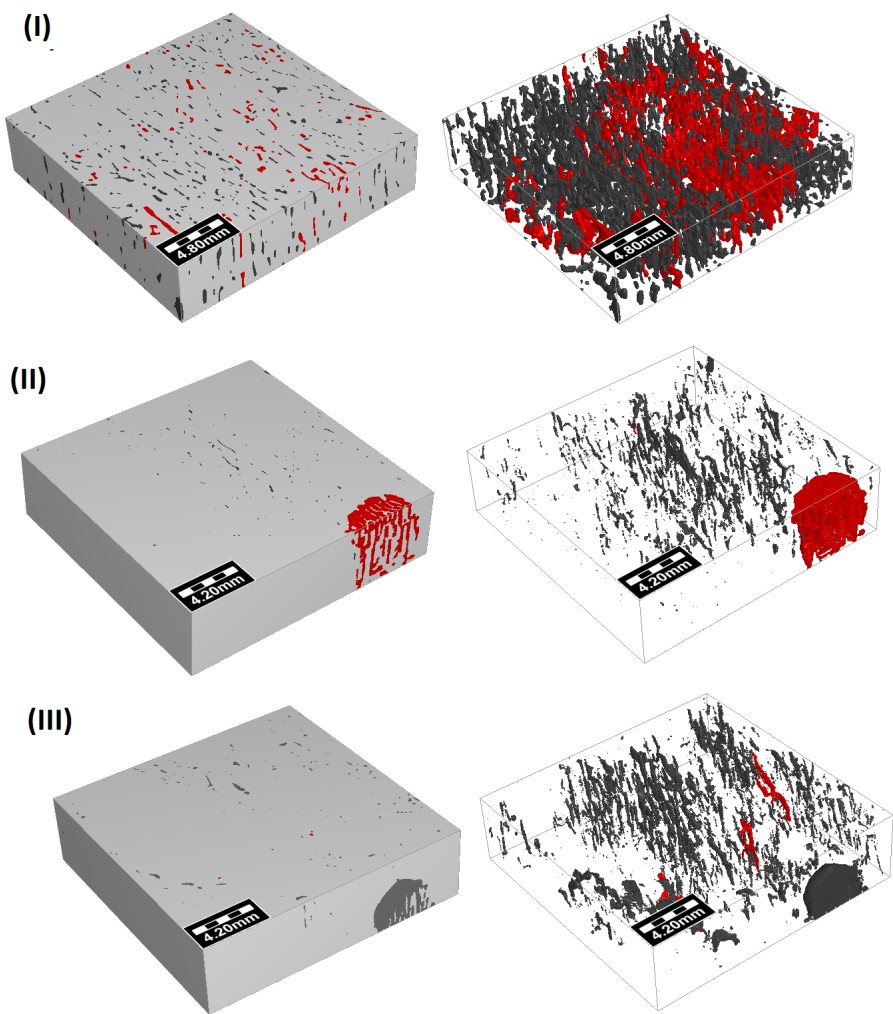

**Figure 8.** 3D images of typical samples as discussed in the text. The left images are 3D images emphasizing the pores visible at lateral boundaries, the corresponding right images show the pore space with ice being invisible, focusing on the sample interior. Vertically connected pores, contributing to the permeability, are shown i red, other pores in dark grey, ice in light grey. I) Only small pores, half of which are connected; II) One connected large pore, no small ones; III) a few connected small pores and one unconnected large pore running out laterally.

The tortuosity shown in Figure 9b is simply the ratio of the length of the maximum diameter path $L_{pth}$ and the sample thickness $L$. For this property no measurable change with porosity is observed, indicating that its influence on the permeability can be considered as small.

### 3.4 Characteristic pore scales

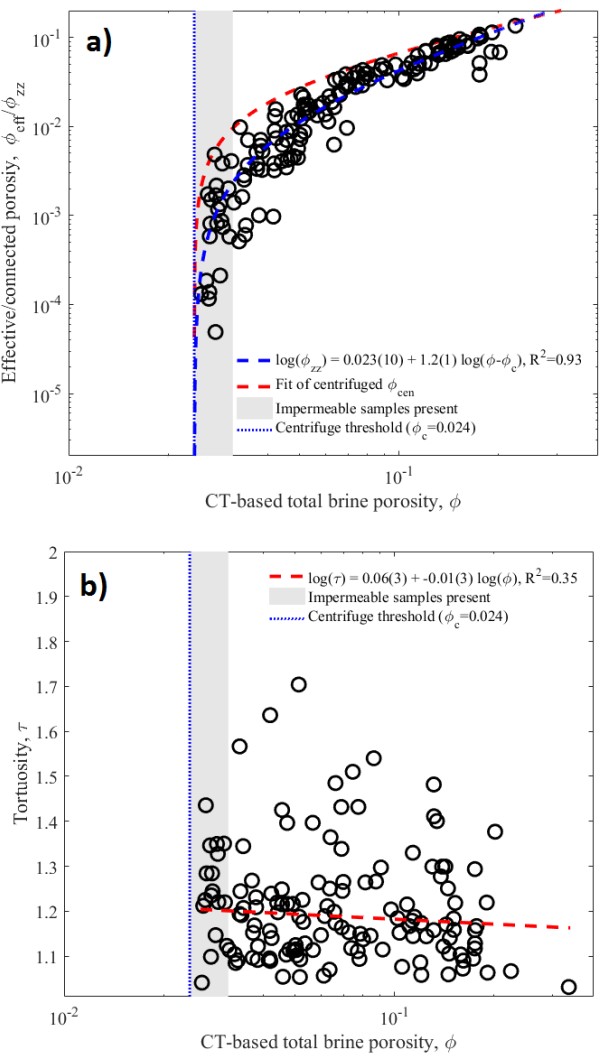

**Figure 9.** a) Relationship between CT-based connected porosity $\phi_{zz}$ and total brine porosity, in comparison to the centrifuge based fit of the open porosity $\phi_{cen}$. b) Tortuosity of the flow based on the length of the path of the channel with maximum diameter.

Average pore size distributions of our data are shown in Figure 10, emphasizing the pore size change during cooling. The left hand figure shows results for open brine pores. The distribution for the two warmest cores with temperature -2 to -4 °C in Figure 3 is shown with red bars, the distribution for the two coldest cores with temperatures -6 to -10 °C with blue bars. The corresponding cumulative distributions are shown as dashed (warm) and dotted (cold) lines. The left y-axis refers to the bars and gives the fraction of open pores in each size class, while the right hand y-axis refers to the cumulative fraction. it is seen that, for the warm and cold sample populations, more than 95% of the pores have a diameter of less than 1 mm. Relative changes due to temperature are largest below 0.4 mm. The median of the open pore diameter, given by the fraction 0.5 in the

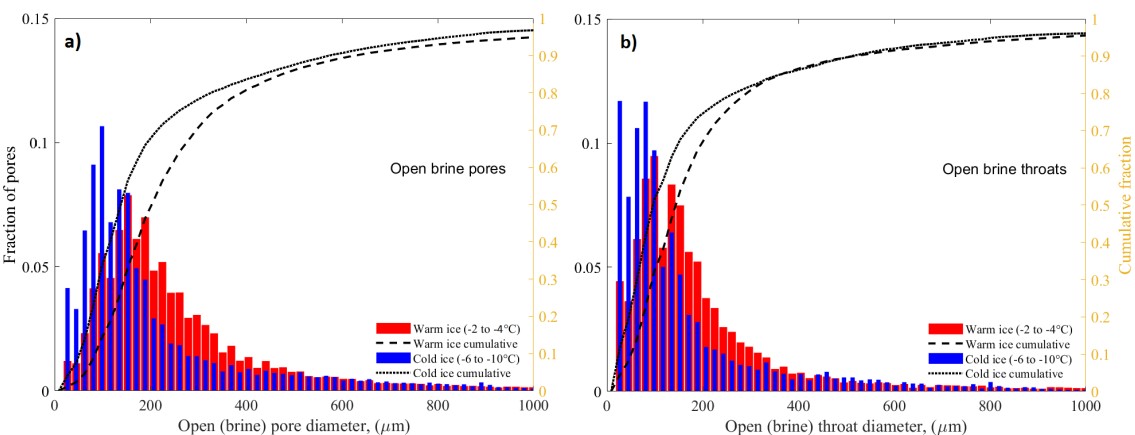

**Figure 10.** New Figure 10 Pore size distributions based on XRT imaging of 4 young ice cores. a) Fraction of open brine pores in 18 $\mu$m wide size bins for the two warmest (red) and two coldest (blue) cores. The corresponding cumulative fractions are also shown, with y-axis on the right hand side; b) same as a) but for the porosimetry/ fraction of pores throats.

cumulative distribution, changes from 0.20 mm for the warm ice to 0.16 mm for the cold ice. Note that the distribution for both warm and cold ice has two modes, one near 0.15 mm and another one near 0.10 mm. The throat size distribution, is similar to the open brine pore distribution with slightly smaller median values of 0.14 mm and 0.10 mm for the warm and cold ice cores, and modes near 0.14 ad 0.08 mm.

Due to variability also in pore scales, the pore size characteristic scales have been determined as median rather than mean values of the volumetric pore size distribution. Figure 11a to d shows their dependence on brine porosity, and that all pore sizes are increasing with $\phi$. This increase has been evaluated by a robust double-logarithmic least squares fit. in order to obtain the power law behaviour of the form $D \sim \phi^e$. This relationship is indicated in the figures. Also shown is the transition regime for which both permeable and impermeable samples have been observed, with grey shading, and a number indicates which scale

the fitted power law has reached at the percolation limit of $\phi_c = 0.024$. A green horizontal line marks the length scale of two voxels (36 $\mu$m) that often is considered as the Nyquist criterion of digital imaging, which states that the sampling interval has to be at least twice the highest spatial frequency to accurately preserve the spatial resolution. This is of particular importance in our study for identification of channels with a path through the sample. A path of just 1 voxel width would very likely be terminated at some level.

The results for the open pore size $D_{opn}$ are shown in Figure 11a. Despite a few outliers, weighted less in the robust fit applied, there is a well-defined relationship with a linear slope in log-log space and $D_{opn} \sim \phi^{0.34}$, with $R^2 = 0.80$. Near the percolation threshold a few lower values are seen to drop below the fit.

The results for the throat size $D_{thr}$ are shown in Figure 11b. They follow in principal the behaviour of the open pore size $D_{opn}$, yet being typically 1.2-1.7 times smaller, and with a slightly steeper slope $D_{thr} \sim \phi^{0.46}$. Also the throat size shows a

drop of a few samples close to the percolation threshold.

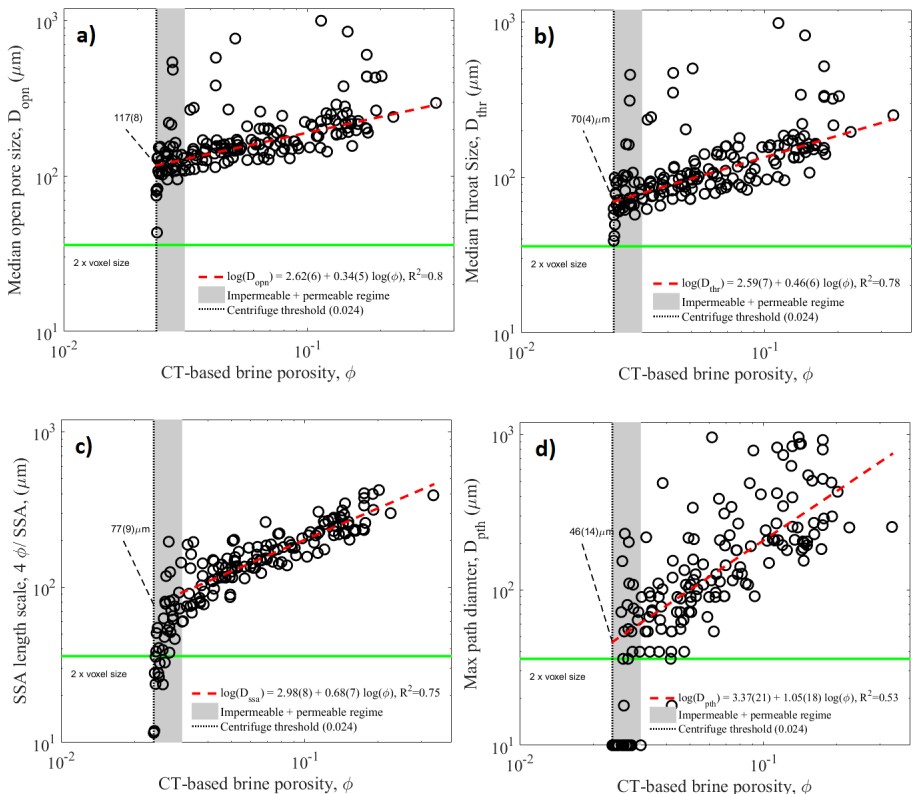

**Figure 11.** Characteristic pore scales and their dependence on brine porosity. a) Median open pore size $D_{opn}$; b) Median throat size $D_{thr}$; c) Pore scale based on specific surface area $D_{ssa}$; d) maximum path diameter $D_{pth}$. For all length scales a power law $D \sim \phi^d$ has been determined by double logarithmic least square fit, shown as red dashed line. The fit is given in the legend, with numbers in brackets giving the uncertainties in the last decimal. For $D_{ssa}$ the fit is only based on data with $\phi > 0.031$, outside the regime where impermeable samples are found. The critical pore scales are obtained where the red dashed lines cut the percolation threshold $\phi_c = 0.024$. The Nyquist criterion (2 x voxel size) is shown as a green horizontal line.

The specific surface area based length scale $D_{ssa}$ is shown in Figure 11c. Here, a linear fit in log-log space obviously does not work for porosities smaller than 0.03, and the grey shaded transition regime has thus been excluded from the fit. The transition of $D_{ssa}$ to lower values then the least square fit (related to larger specific surface) starts at higher porosity than for $D_{opn}$ and $D_{thr}$. That values drop below the proposed resolution limit is related to an algorithm in GeoDict employing estimates of specific surface more complex than a simple sphere-fitting approach.

The last length scale to be considered is the maximum path diameter $D_{pth}$, corresponding to the maximum diameter of a sphere that can permeate through the sample. $D_{pth}$ thus is based on a comparable approach as the throat size $D_{thr}$. The values are much more scattered than $D_{thr}$, and their relationship with $\phi$ has a larger slope with $D_{pth} \sim \phi^{1.05}$, though with less

**Table 3.** Pore scale exponents and thresholds

| Pore scale D | exponent $e$ in $D \sim \phi^e$ | $R^2$ | D at $\phi_c$ |
|---|---|---|---|
| $D_{opn}$ | $0.34 \pm 0.05$ | 0.80 | $117 \pm 8 \mu m$ |
| $D_{thr}$ | $0.46 \pm 0.06$ | 0.78 | $70 \pm 4 \mu m$ |
| $D_{ssa}$ | $0.68 \pm 0.07$ | 0.75 | $77 \pm 9 \mu m$ |
| $D_{opn}$ | $1.05 \pm 0.18$ | 0.53 | $46 \pm 14 \mu m$ |

confidence than for the other length scales. It is seen that the lowest values of $D_{pth}$ are close to the resolution limit line below
a porosity of $\phi < 0.05$.

The exponents of the pore scale versus brine porosity relationships as well as the pore sizes at the threshold porosity $\phi_c = 0.024$ are summarised in Table 3.

## 4    Discussion

We have obtained results for the permeability and pore scales of sea ice through a challenging procedure with the following
steps. (i) Field sampling of a large number of cores (15) of uniform ice, (ii) thorough temperature control of samples at in
situ values, (iii) centrifuging samples at in situ temperatures, (iv) X-ray microtomographic imaging, (v) pores size analysis
and numerical permeability simulations. Also, by lowering the temperatures of harvested ice cores in the lab, we extended the
original *in situ* temperature regime of the samples (minimum -3°C) down to -10°C) and obtained results for brine porosities
down $\phi \approx 0.03$.

It needs to be pointed out that the centrifugation approach has been essential to obtain the XRT results. XRT imaging, the
method of choice for non-invasive imaging of the internal structure of materials (Kinney and Nichols, 1992; Buffiere et al.,
2010), is these days increasingly used in the geosciences (Cnudde and Boone, 2013). It has become an important method in
snow research (Flin et al., 2004; Schneebeli and Sokratov, 2004; Heggli et al., 2011) and recent work has indicated its potential
for sea ice microstructure analysis (Golden et al., 2007; Pringle et al., 2009; Obbard et al., 2009; Maus et al., 2015; Crabeck
et al., 2016; Lieb-Lappen et al., 2017). However a limitation for application to sea ice stems from the small X-ray absorption
contrast between ice and (sea)water (Bartels-Rausch et al., 2014). Imaging sea ice at lower temperature than in the field gives,
due to the corresponding higher salinity of brine, reasonable contrast (Obbard et al., 2009; Lieb-Lappen et al., 2017), yet pore
sizes and connectivity will differ from in situ conditions (as clearly shown in the results presented here). XRT imaging has thus
been performed on ice grown from salt-water with CsCl added as contrast agent (Golden et al., 2007; Pringle et al., 2009). Such
"doping" is not feasible in the field. In the present work, to solve the contrast problem and obtain good images of relatively
warm sea ice, the ice samples were thus centrifuged prior to imaging, replacing brine by air with much higher contrast to ice
(Weissenberger et al., 1992; Maus et al., 2011, 2015).

## 4.1 Effective versus total porosity

Centrifuging is not only a means of obtaining high quality XRT microstructure images. It provides the dependence of centrifuged (effective) porosity on total brine porosity, as well as a porosity threshold of $\phi_c = 2.4 \pm 0.3\%$. This threshold is a new result compared to most earlier work that has more or less accepted a value of $5\%$ (Golden et al., 1998; Cox and Weeks, 1988; Petrich et al., 2006; Golden et al., 2007; Pringle et al., 2009, e.g.,), which will be further analysed below. The derived empirical relationship between effective and total brine porosity, Equation 6, should be relevant for model applications that need to know the effective porosity. The deduced critical exponent $0.83 \pm 0.03$ is of relevance for model approaches based on percolation theory. In terms of the latter $\phi_{cen}$ can be interpreted as the probability to belong to the infinite connected cluster. So far sea ice permeability has been studied in terms of *isotropic percolation* (Petrich et al., 2006; Golden et al., 2007; Pringle et al., 2009), for which the critical exponent for the infinite cluster strength is known to be $\beta \approx 0.41$ in 3D (Stauffer and Aharony, 1992; Sahimi, 1993). However, in sea ice the growth, pore structure evolution and desalination processes are anisotropic and directed towards the ocean. For such a setting, typical for many natural porous media, already Broadbent and Hammersley (1957) have suggested that the percolation should be directed. *Directed percolation* belongs to a different universality class with critical exponents differing from the isotropic case, $\beta \simeq 0.82$ being the presently accepted value for $\beta$ in 3(+1, the direction) dimensions (Henkel et al., 2008; Hinrichsen, 2009). Our deduced $\beta \approx 0.83 \pm 0.03$ is in close agreement with the latter. On the one hand this gives us strong confidence for the validity of the centrifugation approach and its results. On the other hand it points to the need to analyse sea ice in terms of *directed* rather than *isotropic* percolation. E.g., it will be a future challenge to study the anisotropy in permeability, observed by Freitag (1999), and the directional dependence of the porosity threshold, found by Pringle et al. (2009), in terms of *directed percolation*.

We have considered and avoided several possibilities how centrifugation might bias the results. Incomplete centrifugation of brine might lead to brine remnants which, after cooling and freezing, might block pores (Freitag, 1999). This might create a higher apparent porosity threshold, indicated by an earlier study (Buettner, 2011) with lower centrifuge acceleration (15 g compared to our 40 g). By carefully choosing the parameters we think that we largely avoided this problem. Also the warming of ice samples in the centrifuge was carefully tested and avoided by using a centrifuge start temperature 1 K below the in-situ sea ice value. Other effects, like pressure melting of ice or internal deformation, are unlikely at the relatively small centrifuge acceleration rates we used. We cannot exclude that centrifuging has implied minor deviations from in situ temperatures. However, what we derive, in essence and for the first time, from centrifuging and CT imaging, is the relationship between open porosity, total porosity and permeability. We rate it as unlikely that small internal structure changes due to fluid redistribution and freezing/melting in the centrifuge will change this relationship fundamentally.

## 4.2 Effective versus connected porosity

The comparison of CT-based connected porosity $\phi_{zz}$ and open porosity $\phi_{opn}$ to in Figure 9a indicates an increasing difference the lower the total brine porosity $\phi$. The exponent in $\phi_{zz} \sim (\phi - \phi_c)^b$ is $b = 1.2 \pm 0.1$ compared to the exponent $0.83 \pm 0.03$ for the open/centrifuged porosity.

We can obtain a simple estimate of the fraction of brine channels that can be expected to open to the sides and not contributing to $\phi_{zz}$. Assuming a simple 2D geometry and the all pores are parallel, this fraction will be approximately $tan(\alpha)\epsilon$, where $\alpha$ is the inclination angle of crystals/channels against the vertical and $\epsilon$ the ratio of sample height to diameter. For sea ice a typical $\alpha \approx 10°$ has been documented (Kovacs and Morey, 1978; Langhorne and Robinson, 1986; Kawamura, 1988). Freitag (1999) has performed a sensitivity test and found a permeability reduction with sample height that was consistent with $\alpha \approx 10°$. For our $\epsilon \approx 1/4$, the effect is an underestimate by less than 5 %. In the standard experiments from Freitag (1999), with $\epsilon \approx 2/3$, one would expect a slightly larger underestimate of 12 %.

From this consideration we conclude that the inclination of crystals alone cannot explain the increasing difference between $\phi_{zz}$ and open porosity $\phi_{opn}$. There must be operating a pore splitting mechanism that disconnects vertical pores that still are connected to the lateral sides, contributing to $\phi_{opn}$ and not $\phi_{zz}$.

## 4.3  Pore size threshold

In Figure 11a to d we have shown that all characteristic length scales decrease with decreasing porosity. For two length scales, the median open pore size $D_{opn}$ and the median throat size $D_{thr}$, very robust power law relationships of type $D \sim \phi^d$ were obtained. These relationships do not show percolation behaviour of the form $D \sim (\phi - \phi_c)^e$, but they are supposed to create the percolation behaviour in $\phi_{zz}$ and $\phi_{opn}$ as follows. By evaluating the power law relationships at the present percolation threshold $\phi_c$, we obtain their critical values at the percolation threshold. Of particular interest is the critical throat diameter

$$D_{thr,c} = 70 \pm 4 \ \ \mu m, \ \ at \ \phi_c = 0.024 \tag{9}$$

at the threshold. We interpret it as the throat diameter at which necking occurs to lock the brine pores.

This result is consistent with two earlier studies of sea ice microstructure. Anderson and Weeks (1958) discussed the transition from brine layers into cylindrical brine tubes in connection with changes in the relationship between sea ice strength and brine porosity. They proposed, based on an analysis of horizontal thin sections, a splitting of layers into channels near a tube diameter of 0.07 mm. These authors have not presented a statistical analysis of their results, but mention that they obtained the value from 'photographs of layers just before and after the splitting'. From the plate spacing reported for their study ($0.46mm$ on average) it may be suspected that they analysed mostly young ice of similar age than ours. The agreement of ours and their result is very interesting. Also Light et al. (2003) studied the temperature dependence of sea ice microstructure, in order to formulate a model for the radiative properties of sea ice. Based on the optical analysis of many samples they distinguished morphologically between brine tubes (above a length of 0.5 mm) and brine pockets (below this value), and derived an equation for the aspect ratio (length $L$ divided by diameter $D$) of tubes and pockets ($10.3D = L^{0.33}$). Inserting the pocket-tube transition of 0.5 mm for $L$ one obtains a tube diameter of 0.077 mm at the transition, indicating also here a similar scale for the splitting of tubes.

The critical median open pore size $D_{opn,c}$ computed at the threshold was $117\mu$m, a factor of 1.7 larger than $D_{thr,c}$. This is likely the value that one would identify by considering all pores in a 2D thin section, because one would not know which are the throats. Also noteworthy, though not investigating the temperature dependence or transition from pockets to tubes, is

the study by Cole and Shapiro (1998) of Arctic first-year ice at the start, mid and end of the freezing season. These authors were not simply doing 2D thin sections, but sectioned sea ice vertically and horizontally, to obtain the dimensions of brine filaments in three directions (their Figure 9). They found the average width of brine inclusions, at a depth of 0.2 m, to increase from $0.08 \pm 0.03$ mm to $0.14 \pm 0.04$ mm (their Figure 10b). The imaging temperature was -14 °C which, with the reported salinities of 5-7 psu, indicates a porosity of 0.022 to 0.031. This condition is similar to the percolation limit in our study, which is supported by the fact that Cole and Shapiro (1998) indeed mostly observed vertically disconnected brine filaments. The range of observed brine inclusion widths is consistent with our median open pore size $D_{opn,c}$. Also Perovich and Gow (1996) have optically analysed sea ice inclusions in 2D thin sections, focusing however on other microstructure characteristics. From their tabulated values of major axis length, perimeter and circularity of ellipses that were fitted to brine pores, one can deduce a minimum axis length. Median values obtained in this way (see Maus (2007)) fall in the range of 0.05 to 0.1 mm and are comparable to our observed values. However, from the 2D data no information on pore connectivity and necking is available.

The analysis of pore and throat diameters thus gives us important information about the critical length scales at the percolation transition. More supporting information comes from the specific surface area length scale that we compute by assuming that the surface area relates to infinite pores with circular cross section, which means $D_{SSA} = 4\phi/SSA$. This is the only length scale that appears to show critical behaviour near the percolation threshold. This behaviour indeed supports the necking hypothesis as follows: Consider a long brine pore that splits into spherical inclusions. While $D_{opn}$ and $D_{thr}$ will not change much, the SSA does increase during the transition to spheres. However, to account for this in the length scale computation one would have to calculate $D_{SSA} = 6\phi/SSA$. As this is not done for the data points in Figure 11c, there is an apparent drop in our computed $D_{ssa}$ when splitting takes place, nicely seen in our data.

But we can, through $D_{ssa}$, not only identify the necking and splitting near the percolation threshold. When considering the power law fits $D \sim \phi^d$ one would expect that, if decreasing $D_{SSA}$ with $\phi$ would only relate to diameter changes, it should be described by a similar exponent $d$ as $D_{opn}$ ($0.34 \pm 0.05$) and $D_{thr}$ ($0.046 \pm 0.06$). However, also if the fit is restricted to the regime $\phi > 0.031$ we find an exponent (equivalent to the slope in log-log space) that is larger for $D_{ssa}$ ($0.68 \pm 0.07$). The interpretation is that slitting and necking operates over the whole porosity regime in our dataset.

The critical value of $D_{thr,c} \approx 0.07$ mm should probably be interpreted as a statistical descriptor of the pore space, rather than a strict limit. Looking at the fourth characteristic length scale, the maximum path diameter $D_{pth}$ in Figure 11d, we see that there exist through-flow paths with lower diameter. This is not unexpected in the sense that the throat size distribution only has its median at $70\mu$m at the transition. Another approach to estimate the critical $D_{thr,c}$ is based on the following argument. Cooling ice does decrease the brine volume and leads to shrinking of pores and, for a range of ice temperatures, to a broad distribution of pore sizes. If however there is a preferred pore size for necking than pores will not shrink around this value, as internal freezing now rather closes pores. Hence, one would expect a local maximum in the pore size distribution. A look at average throat size distribution in Figure 10b indeed shows such a maximum. For the cold ice (blue bars) it is located near 0.08 mm (in the size class 72 to 90 $\mu$m) and hence consistent with the result from the least square fit. For a more detailed discussion the present dataset is somewhat limited here, as the lower range of the identified maximum path diameters touches

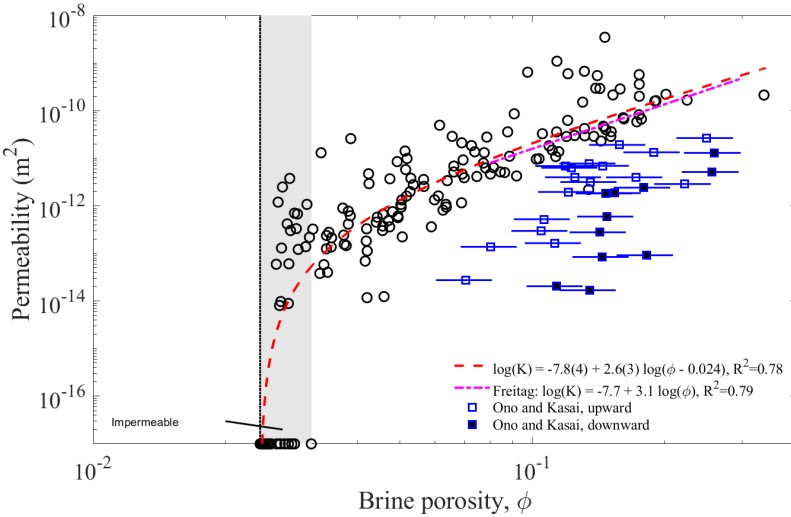

**Figure 12.** Relationship of simulated vertical permeability $K$ and $(\phi - \phi_c)$ as shown in Figure 7 as red dashed curve, here compared to two earlier investigations. As the measurements from Freitag (1999) are only available in terms of $\phi_{cent}$, we plot the least square fit from Freitag (cyan curve, numbers given in the legend). The data points from Ono and Kasai (1985) represent downward and upward permeation of brine through the ice - see text for more information.

the Nyquist spatial criterion of $36\mu$m below a porosity of $0.05$. Figure 11d indicates that, to study the necking transition near the percolation threshold dynamically, one would likely have to increase the present resolution by at least a factor of two.

Regarding the mechanism of necking, Anderson and Weeks (1958) had once argued that the necking of pores is driven by surface energy effects. However, one of us has argued that (i) the original brine layers are expected to be low energy surfaces and that (ii) latent heat energy fluxes during freezing are many orders of magnitude larger than surface energy transitions (Maus, 2007). Due to these factors it seems more likely that morphological freezing instabilities in supercooled brine layers and pores play a role for the necking, in a similar way as they do for the plate or brine layer spacing at the ice-water interface

(Wettlaufer, 1992; Maus, 2020). A concise physical explanation for the necking phenomenon is lacking so far. Progress could be made by direct observations of the 3D pore space evolution by X-ray tomography, building on 2D visual observations of pore necking described by Niedrauer and Martin (1979) for a thin growth cell. Such an approach may be feasible through fast time-lapse synchrotron-based X-ray tomography for low contrast materials (Beckmann et al., 2007; Buffiere et al., 2010, e.g.,). Also conventional laboratory-based XRT with higher spatio-temporal resolution may provide new insight into the details of

necking and pore instabilities.

### 4.4   Permeability

Having found consistent explanations of the observed percolation limit in terms of critical pores sizes, we now return to the permeability simulations. In Figure 12 we compare our results to the results of two experimental studies on young ice.

The relationship based on the work by Freitag (1999) is likely the most frequently cited and used in the literature. Freitag only documented the data points of $K$ versus the centrifuged brine porosity $\phi_{cen}$, yet he has given the relationship between $K$ and total brine porosity $\phi$. It is given in the legend of Figure 12 and plotted as a green dashed curve. The porosity range for Freitag's fit ($0.07 < \phi < 0.3$) has been estimated from the $\phi_{cen}$ values he reported. Comparison with our fit (the red dashed curve) indicates a very good agreement with Freitag's results for his porosity range of validity. The maximum difference is just $30\%$. As the confidence bounds for our fit indicate roughly a factor of 2, there is no significant difference in the predictions. However, our relationship is based on a fit of observations down to $\phi = 0.03$, and allows to estimate permeabilities at low porosities, where Freitag's relationship is not applicable.

The second dataset is from experiments that Ono and Kasai (1985) performed with artificially grown young ice. These authors did not publish the permeability but the hydraulic conductivity, and we have used Equation 3 to convert from $\overline{V}$ to permeability $K$. To do so we used the temperature and brine salinity dependent kinematic viscosity relationship from Maus (2007). As Ono and Kasai only reported surface temperatures of their tested ice, we make the assumption that the salinity is similar to those reported for ice growth experiments from other laboratory studies. For the documented ice growth velocity of $10^{-4}$ cm/s and water salinity ($S_w \approx 33\ \text{\textperthousand}o$ NaCl) a 6 cm thick sea ice crust will typically contain $40$ to $50\%$ of the salinity of water from which it grows (Wakatsuchi, 1974, 1983). We thus assume a sea ice salinity of $S_i \approx 14 \pm 2\ \text{\textperthousand}o$ NaCl. We shall use this range with reported surface temperatures to estimate the brine volume $\phi$ at the surface of the ice. This leads to the blue data points and uncertainty estimates in Figure 12. There are two sets of data. The open squares are based on measurements of upward movement, where Ono and Kasai created a pressure gradient directed from the water into the ice. The filled squares are based on measurements of downward movement, where the authors poured brine, in salinity equilibrium with the surface temperature, onto the ice. The surface temperature was adjusted with an infrared lamp.

It is seen that the permeability data from Ono and Kasai (1985) fall 1-2 orders of magnitude below our and Freitag's observations. The error bounds indicate that this hardly can be explained by a lower salinity of freshly grown ice (at $10^{-4}$ cm/s) than assumed. We think that the difference is related to two factors. The first is related to the fact that, in contrast to our and Freitag's study of sub sample permeability, Ono and Kasai measured the permeability of the full ice thickness (of however only 6 cm). In this setting the ice surface, where the ice is coldest, will control the permeability. However, in most cases the first surface ice skim is growing much faster, implying smaller crystals (or plate spacing) and thus more tiny pores between them, with strong impact on the permeability (e.g., Okada et al., 1999). In addition to crystal size a more random crystal orientation may imply tortuous flow and decrease the permeability further. The second factor is likely related to changes in the microstructure during the experiments, also discussed by Freitag (1999). Assume that the upward flow experiments were performed first. As during the flow less saline seawater is exchanged against brine, the ice salinity will decrease. At high surface temperatures (similar to those in the seawater) this salinity decrease is low, yet it becomes large with the temperature difference between the ice surface and the water. This is consistent with larger deviations of the data points from Ono and Kasai from our permeability fit at lower brine porosity. If the downward flow experiments were than to take place after the upward flow, salinities could have been considerably less than assumed for freshly grown ice. The argument also works vice versa, to explain the difference in the upward and downward results, as during downward flow of high salinity brine the bulk

salinity of an ice sample is expected to increase. Based on these arguments, the data from Ono and Kasai (1985) provide some qualitative support for the existence of a sharp transition in permeability at some critical brine volume fraction or temperature, as proposed by Golden et al. (1998). However, the nature of their experiment was not suited to validate such a brine volume threshold quantitatively.

We recall that the double-logarithmic fit in Figure 12 is only based on our data above $\phi = 0.031$, excluding the regime where we find permeable and impermeable samples. The investigation of the characteristic pore scales, in particular the maximum path diameter shown in Figure 11d, also indicates that we start facing resolution problems below a brine porosity of $\phi \approx 0.04$. This, as well as image segmentation errors, may to some degree explain that the data points in the regime $0.024 < \phi < 0.031$ appear significantly above our proposed relationship. And one can argue, that some kind of averaging of zero permeabilities with these high values would move the data closer to the fitted percolation curve. As our data in this regime are not of high enough quality we have not applied such a correction. However, the combination of good enough image quality above $\phi \approx 0.04$ has, in combination with the threshold $\phi_c = 0.024$ from the centrifuge experiments, enabled us to deduce Equation 8 with good confidence, at the same time being consistent with the work from Freitag (1999).

Over the whole range of porosities, we find some samples with O(2) larger permeabilities than the fitted curve and most data points. As discussed above these are associated with type (II) ice samples in Figure 8 and the presence of larger brine channels. Such a dual system of pore sizes is a frequently described, yet but in detail still little explored, feature of sea ice (e.g., Wakatsuchi, 1983; Wettlaufer et al., 1997; Weeks, 2010; Rees Jones and Grae Worster, 2014). That only part of our samples do contain larger brine channels is likely related to our limited sample size (3 cm diameter resulting in 2 cm side length for numerical simulations), in turn leading to the scatter in permeabilities. However, the large number of samples, and the additional constraint $\phi_c = 0.024$ from centrifuging larger samples, has allowed us to obtain a statistically robust relationship between $K$ and $\phi$. Yet the uncertainty is still O(100%). Improvements may be made on the basis of currently available X-ray detectors that allow for a 2 times larger field of view (at the same spatial resolution), with a better representation of the dual brine pore size networks.

For modelling efforts of the permeability with porosity, the obtained exponents are of high relevance. In general the dependence of permeability $K$ on porosity $\phi$ is often empirically characterised by an equation of the form $K \sim \phi^b$. The simplest model is to relate porosity $\phi$ to pore diameter $d$ and assume the well known relationship $K \sim \phi d^2$. From the basic sea ice microstructure of parallel brine layers ($d \sim \phi$) and circular tubes ($d \sim \phi^{1/2}$) one could argue for $2 < b < 3$. The larger exponent $b = 4.0 \pm 0.4$ that we find (Figure 7) reflects that not only pore diameters are changing with porosity but also their connectivity. Percolation theory accounts for this effect, which is resembled by an equation that involves the threshold porosity in the form $K \sim (\phi - \phi_c)^t$ (see also Eq. 8). For isotropic percolation this has been first proposed as 'critical path analysis' for the electrical conductivity (Ambegaokar et al., 1971), and the current best estimate for the conductivity exponent, in the form $E \sim (\phi - \phi_c)^e$ is $e = 2.0$ (Stauffer and Aharony, 1992). The permeability exponent $t$ however will be larger than $e = 2.0$ if the characteristic pore scale $d_c$ of permeating paths is changing with porosity. This problem has been addressed by several authors proposing an equation of the form $K \sim d_c^2(\phi - \phi_c)^e$. E.g., Katz and Thompson (1986) proposed to determine the length scale $d_c$ on the basis of mercury porosimetry. This is indeed the approach (virtual porosimetry) we have applied to determine the throat sizes in the

present study. If we insert the throat size exponent from Fig. 10b, $d_c \sim \phi^{0.46}$, and assume $e = 2.0$ we would get a dependence

of the form $K \sim \phi^{0.92}(\phi - \phi_c)^2$. This in turn is not very different from our percolation-based fit $K \sim (\phi - \phi_c)^{2.6}$. We note that Golden et al. (2007) have proposed $t = 2.0$ for older sea ice, arguing that wide brine channels with a constant critical diameter control the permeability, and also reported $t = 1.97$ as a best fit to the data from Ono and Kasai (1985). We were unable to reproduce the latter result based on the data in Figure 12, using our estimates of ice salinities in the experiments from Ono and Kasai (1985). Our best fit for the permeability critical exponent, $t = 2.6 \pm 0.3$, is largely consistent with percolation theory,

critical path restriction by throats and their dependence on porosity. It is valid for young sea ice where the permeating pores are shrinking with decreasing porosity, and will differ for ice with a different pore-porosity relationship. However, also for young ice a general prediction may be more complicated due to three aspects: First, it has been pointed out by Le Doussal (1989) that the approach from Katz and Thompson (1986) needs to be revised for broad pore size distribution which may lead to higher exponents for the permeability. Second, the exponents may also be different for directed percolation. And third, ice type may

play a role and results for granular ice may be different. We are currently investigating this problem in more detail.

## 4.5 Porosity threshold

The present analysis has enabled us to deduce a porosity threshold of $2.0 < \phi_c < 2.9\%$. This optimal threshold cannot be deduced from the CT measurements alone as these data are scattered, and the CT samples are 1/50 in volume of the centrifuged samples. However, the evidence based on the larger centrifuged samples is much stronger. Our confidence is related to the power

law fit of the centrifuged porosity in Figure 4 (critical exponent $\beta = 0.83 \pm 0.03$) and its consistency with the theoretical critical exponent from directed percolation (critical exponent $\beta \simeq 0.82$). Based on this agreement we can state that, if our hypothesis is correct, that the pore space evolution of sea ice follows the behaviour of *directed percolation*, then this implies a threshold porosity for percolation in the range 2 to 3 %. The CT-based microstructure analysis supports these results, and further can be interpreted in the way, that the threshold is related to the necking or close-off of pores at a critical diameter of 0.07 mm. Our

deduced porosity threshold is just half of the value of $\phi_c = 5\%$ once proposed by Golden et al. (1998), that since then has been confirmed in other studies and become the mostly accepted threshold for sea ice permeability and desalination (Weeks, 2010). In the following we discuss these studies in the context of our results.

The first proposal of a critical brine porosity and permeability of sea ice was once published by Golden et al. (1998). These authors proposed that sea ice typically becomes impermeable when its salinity is 5 ppt, its temperature -5°C, and its brine

porosity 5%, which is now known as the 'rule of the fives'. To support this hypothesis the authors used a percolation model analogy of compressed powders, where the threshold depends on the ratio of critical brine inclusion to ice crystal thickness scales. As experimental evidence the experiments from Cox and Weeks (1975) and Ono and Kasai (1985) discussed above were considered, indicating a porosity threshold of 5% associated with a temperature -5 °C. However, on the one hand the model is simplistic and not backed up by detailed microstructure observations. On the other hand, our analysis above indicates that the

experiments from Ono and Kasai (1985) are difficult to interpret. Assuming typical young ice salinities, the experimental data from Ono and Kasai (1985) would be far away from a porosity of $\phi \approx 5\%$, while the suspected changes in ice salinity during

the the experiments are unknown. Due to these considerations, and comparison to our simulations in Figure 12, we think that the data points from Ono and Kasai (1985) cannot be used to qualitatively constrain a percolation threshold.

An earlier proposal of a critical brine porosity $\phi_c = 5\%$ has once been suggested by Cox and Weeks (1988), based on observations of observed salt fluxes from sea ice (Cox and Weeks, 1975). The data has been later analysed in more detail by Petrich et al. (2006), coming to the conclusion that sea ice permeability limited to brine porosities above $\phi_c \approx 5.4\%$. However, there is a general problem with this argument: It is not the vertical permeability that has been observed by Cox and Weeks (1988), but the desalination of the ice. The latter however depends on other factors, like the brine salinity gradient in the ice and as well as the horizontal permeability to drive internal flow. The analysis may thus be interpreted to represent a porosity threshold at which convection sets in, rather than at which the ice becomes impermeable.

The most stringent approach to estimate $\phi_c$ was proposed by Pringle et al. (2009), based on the first 3D analysis of CT images. These authors focused on the vertical connectivity $\phi_{zz}$ and investigated its scale dependence to estimate the connectivity threshold based on assumptions from isotropic percolation theory (Stauffer and Aharony, 1992). They investigated artificial sea ice images, cubic and with side lengths 2 to 7 mm. From the scale dependence of $\phi_{zz}$ they deduced a critical value of $\phi_c = 4.6 \pm 0.7$ % for the vertical percolation threshold. This result thus seemed to support the earlier work and 'rule of the fives'. However, in view of the present study and in particular the throat size threshold, also this result may need some revision. The critical aspect is that the detectability of pores was likely limited to widths of $83 \mu$m (Nyquist criterion of two times the voxel size). In our study, with a $36 \mu$m Nyquist criterion, we are observing larger scatter in connectivity and pore scales, when the porosity threshold is approached. We thus suppose that such problem may have influenced the percolation behaviour of the samples from Pringle et al. (2009) at two times coarser resolution. E.g., considering our deduced critical throat size of $70 \mu$m, a similar value would not have been resolved by imagery with a $83 \mu$m Nyquist criterion. A simplistic quantitative argument may be obtained by looking at Figures 11b for the throat size $D_{thr}$ and 11d for the maximum path diameter $D_{thr}$. We can ask at which porosity the lowest observed median throat sizes drop below $83 \mu$m, which indeed happens in the range $5 < \phi < 6$ %. Finally, though likely of minor importance, the results from Pringle et al. (2009) can be expected to change if critical exponents for directed rather than isotropic percolation, supported by the present study, would have been used in the derivation.

In summary, we interpret the earlier work as follows. The proposal of the 'rule of the fives' by Golden et al. (1998) was based on measurements by Cox and Weeks (1975) showing that sea ice desalination due to gravity drainage almost vanishes for brine volumes below 5%, and on observations of the hydraulic conductivity of young ice (Ono and Kasai, 1985) showing large changes over a small temperature range. As the ice in the latter study likely had an initial porosity much larger than 5%, and the data are difficult to interpret, they do not provide evidence for a 5% percolation threshold. While an indirect approach by Petrich et al. (2006), also analysing the desalination data from Cox and Weeks (1975), supported a 5% threshold, we point out that the latter study strictly only applies to the driving force of internal convection, not to the permeability itself. Last but not least, results from a CT-image based study by Pringle et al. (2009) in support of $\phi_c$ of 5% may have been resolution limited (in view of the present higher resolution results). Hence, many studies and datasets of young sea ice that have been proposed earlier in favour of a 5% porosity threshold seem to require a revision, while our confident threshold range of 2 to 3% from centrifuging is a factor of 2 lower. We finally add a note on the question if the true threshold porosity might be even smaller, and

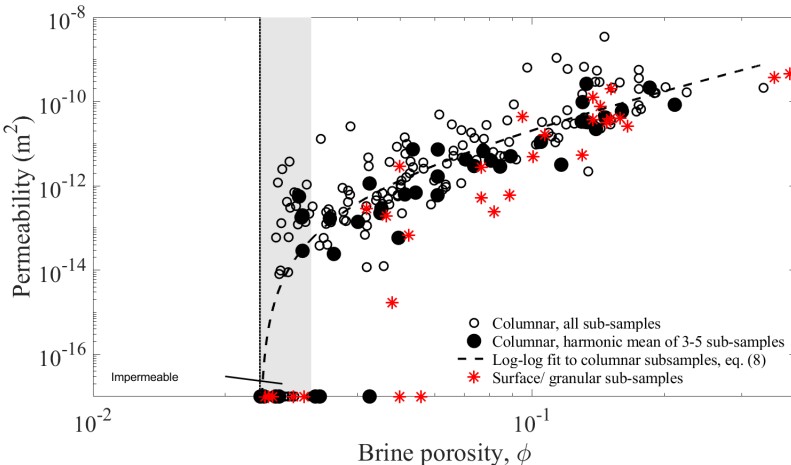

**Figure 13.** New Figure Comparison of simulated vertical permeability $K$ and $(\phi - \phi_c)$ for columnar samples (as shown in Figure 7 and here as open circles) to two other simulation results. The solid circles are harmonic means of all subsamples (normally four to five) from each basic sample cut in the field (and centrifuged). The red stars are permeability results for near-surface samples (up to 5 cm from the surface) that were classified as granular ice and thus excluded from the basic analysis.

was limited by our centrifuge acceleration of 40g. As discussed in connection with equation 4, we estimated that our settings should be valid to retrieve permeabilities as low as $10^{-14}$ m$^2$. Hence, though we may have missed lower values, our overall data are consistent with percolation theory and the here proposed porosity threshold.

### 4.6 Other ice types and growth conditions

4.6. Other ice types and growth conditions The discussion of earlier work, and the present results apply to the permeability of young columnar sea ice during its growth phase, at a stage when mostly primary brine channels and pores exist. Here we discuss possible implications for other ice types, age, thickness, and scale effects. We consider four aspects to be most relevant to generalize our results. These are dependence of permeability on (i) ice growth velocity, (ii) ice type/texture, (iii) ice age and (iv) scale effects due to full thickness finite sample sizes. The first three aspects are related to natural variability in growth conditions and thermal history. The fourth aspect is related to the process to be investigated (e.g. full depth permeability for surface flooding versus near-bottom permeability for desalination/internal convection). It also relates to the question, if tested samples are representative volume elements for the process and represent sea ice macrostructure.

(i) Our range of $2-3\%$ for $\phi_c$ is valid for young ice that has grown at moderate growth rates (2-5 cm/day for both our and Freitags experiments). We conjecture that this threshold is not a constant for sea ice but depends on growth conditions. The basic argument is that, if the critical length scale for necking of throats controls the transition, the critical brine volume $\phi_c$ may be expected to simply scale inversely with the spacing of these throats. Assuming that this spacing is proportional to the basic brine layer or plate spacing $a_0$, one would expect that $\phi_c \sim a_0^{-1}$, implying that the percolation threshold in slower growing ice

(with larger $a_0$) should be smaller. This effect may potentially also explain differences between our results and other studies discussed above, yet would require more data to be proven.

(ii) Sea ice may grow as columnar or granular ice, and the latter ice type often prevails at the surface or the upper centimetres. In our ice cores the upper two samples were granular and these have been excluded from the present pore scale analysis of columnar ice. In Figure 13 we also show results for these granular samples. The number of data points is limited yet seem to indicate a higher porosity threshold. While the small number of granular samples are insufficient for a statistical significant conclusion (contrasting the large number of columnar samples), the larger porosity thresholds reported by Golden et al. (1998) for surface flooding and full depth percolation may be viewed in this context.

(iii) During aging and thermal cycling, sea ice develops wide secondary brine channels systems (Weeks, 2010). These larger pores will then control the permeability that can be orders of magnitude larger. In our young ice there are some wider channels, leading to samples with 1-2 orders of magnitude larger permeability. However, as shown in Figure 10, the majority of the samples is lacking such wider secondary channels, and the permeability is controlled by the primary network. There is in general a lack in data on permeability and pore sizes as well as the the porosity threshold of older sea ice (Freitag, 1999; Freitag and Eicken, 2003; Golden et al., 2007). E.g., sack-hole measurements of permeability reported by Golden et al. (2007) show considerable scatter. It will be a future challenge to determine how secondary channels evolve in time and space, and how this depends on fluid flow and permeability within the finer primary pores.

(iv) Scale effects are related to the question: Which is the length scale of internal fluid flow for which we need to know the effective permeability? In Figure 7 and 12 we have presented our permeability results for samples of vertical extension 5.5 mm. These indicate a scale effect due to finite sample sizes visible in the sample to sample variation of permeability. The reason for this variation is that the frequency of wider brine channels is too low to be presented in all our samples. However, due to the large dataset, and the constraints on $\phi_c$ from the centrifuge-experiments, this effect of finite samples sizes is not critical for our results. In Figure 13 we further compare these results to (harmonic) mean permeabilities of 3-5 sub-samples, corresponding to sample heights of 17-28 mm, and do not find a significant difference in the permeability-porosity relationship. We thus believe that our volumes have been sufficiently large to be interpreted as representative volume elements for young sea ice, also when comparing them to only moderate finite size effects in connectivity reported by Pringle et al. (2009) for 2 to 7 mm sample sizes. The present results should thus be relevant for convection and desalination modelling in the near-bottom regime and skeletal layer of sea ice. Processes like surface flooding and melt pond drainage would depend on the full depth permeability, and thus depend on the lowest local permeability values. This again raises the question, if the percolation threshold of granular ice is given by a higher brine porosity. There is a need for more data.

## 5  Conclusions

Most previous investigations on the permeability of sea ice have been facing challenges related to difficulties to observe this property in situ, and to the reactive nature of sea ice during transport, storage and experiments. The 'rule of the fives' once proposed by Golden et al. (1998) has provided, together with ideas to model sea ice on the basis of percolation theory (Golden

et al., 1998, 2007; Petrich et al., 2006; Pringle et al., 2009), an attractive and reasonable rule of thumb to understand this important sea ice property better. However, much of these earlier results have been based on indirect observations (like surface flooding and desalination) and a stringent validation has been lacking for sea ice and its different micro-structures. First now it has been possible for us, through larger and higher resolution 3D imagery and higher performance computing, to study the pore scale details of sea ice and perform direct numerical simulations of its permeability. We have investigated the percolation behaviour of young Arctic sea ice in terms of the two non-destructive techniques (i) centrifuging of brine for separation of the connected and disconnected pore space and (ii) 3D X-ray microtomographic imaging followed by direct numerical simulations of the permeability and an analysis of the relevant pore size characteristics. Our main findings are

– We obtain a confident relationship between centrifuged (effective) porosity $\phi_{cen}$ and total porosity $\phi$ of young sea ice

  – The relationship $\phi_{cen}(\phi)$ strongly supports that sea ice should be analysed and modelled in terms of *direction percolation* theory, rather than its *isotropic* variant so far applied to sea ice problems.

  – We further find that the relationship $\phi_{cen}(\phi)$ is consistent with a connectivity threshold at a porosity of 2 to 3 %. This value is considerably lower than the commonly accepted $5\%$ based on earlier investigations (Golden et al., 1998; Cox and Weeks, 1988; Petrich et al., 2006; Golden et al., 2007; Pringle et al., 2009).

  – Our pore scale observations near the percolation threshold indicate that earlier estimates of $\phi_c$ were likely limited by a too coarse spatial resolution, or cannot be strictly related to a percolation threshold.

  – We associated the percolation transition with the necking of brine pores, identified it with the median of the critical pore throat diameter distribution, and obtained an estimate of the critical throat diameter or width $D_{thr,c} \approx 0.07$ mm at the transition. This finding is consistent with pore size analysis from earlier studies.

  – We derived a novel consistent parametrisation of vertical permeability $K$ based on total brine porosity $\phi$ that is valid for the porosity range 0 to 20 %, improving and extending earlier work.

The centrifuge approach requires very good logistics, in particular the field operation of several temperature control boxes and a refrigerated centrifuge. Yet it has several other advantages related to the challenge to really image the in situ sea ice microstructure. A well known problem for high porosity samples is brine loss during sampling, which in non-centrifuged samples would show up as a lot of air. By centrifuging all connected brine out, this problem is solved. The connected and open air fractions are deduced by XRT image analysis, and the connected air is associated with connected brine. The approach also considerably increases the image quality, as XRT imaging of brine networks at high temperatures suffers from contrast limitation between ice and brine (Pringle et al., 2009; Bartels-Rausch et al., 2014). Replacing the interconnected brine by air makes not only XRT imaging of the emptied pore networks possible, it also makes microstructure changes during storage and transport much less a problem.

The present work presents new insight into the sea ice pore space evolution of young sea ice and theoretical interpretation of the latter. It demonstrates the large potential of 3D X-ray micro-tomographic imaging to make progress in our fundamental

understanding of sea ice properties. For future work we suggest several directions to make further progress. First of all, the result for permeability-porosity relations and thresholds, are not directly transferable to the thicker, older and warmer summer ice. The latter often contains coarser secondary brine channels that are lacking in young ice and that are relevant for processes like melt pond percolation and melt pond albedo feedbacks (Freitag and Eicken, 2003; Polashenski et al., 2017). This reflects one of the challenges in sea ice physics, which is to improve our understanding how the sea ice pore space, as well as permeability and other physical properties evolve over time. To make progress more 3D CT data of sea ice and its pore space evolution over time are needed. Second, while the present study may be seen as a starting point to a concise understanding and modelling of this evolution, it should be verified with higher spatial resolution to clarify any resolution limit with respect to necking and porosity thresholds. Third, there is a need for comparing granular and columnar ice, as the granular surface layer will be important for the through-flow permeability. And last but not least, due to the lack in experimental data, carefully controlled laboratory experiments in the lines of Ono and Kasai (1985) would be of high value. Combing such experiments with repeated CT imaging to monitor flow-induced microstructure changes, could provide valuable insight about the evolution of the sea ice pore space and its permeability. A useful concept with centrifuged sea ice would be measurements of kerosine permeability in a permeameter (Saito and Ono, 1978; Saeki et al., 1986), allowing validation of permeability computed from CT imagery and the question if there are sub-resolution pathways.

*Data availability.* The cropped microCT-scan data analysed in this study (170 Gigabytes) are made available online by Math2Market (Maus. et al., 2021). The full dataset (670 Gigabytes) will be made available upon publication of this manuscript in the UNINETT Sigma2 research data archive.

*Author contributions.* SM performed the field experiments, the data analysis and writing of the first manuscript. SM performed the X-ray scanning with support from MS and his working group. SM run the permeability simulations and pore space analysis using Geodict with support from AW and Math2Market. All authors contributed to rewriting and submission of the manuscript. All authors improved the manuscript contributed to field data acquisition and contributed to writing of the manuscript.

*Competing interests.* All authors declare no competing interests.

*Acknowledgements.* This project was partly funded through the Research Council of Norway (RCN) program PETROMAKS2 grant 243812 (Microscale Interaction of Oil with Sea Ice for Detection and Environmental Risk Management in Sustainable Operations, MOSIDEO, 20015-2020) as well a the RCN grant 218407 (Microstructure and phase transitions of sea ice, MIPHASICE, 2011-2012).

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
