# Peer review of "An X-ray micro-tomographic study of the pore space, permeability and percolation threshold of young sea ice"

_The Cryosphere, 2020_

## Referee Comment (RC1) · Anonymous Referee #1 · 11 Dec 2020

**1   Summary**

This paper presents a new experimental characterization of the pore space and permeability of natural sea ice. The techniques are advanced and novel for natural sea ice. These measurements have a wide importance to those studying the evolution of sea ice, since many processes are very sensitive to permeability. The experimental technique is well described and careful. The results are presented clearly and provide strong evidence in contrast to widely-cited previous studies showing a percolation threshold at 5% porosity. The limitations of the study are discussed very well, although there are three significant areas in which I think limitations need more discussion. The

writing quality is excellent. Overall, I think the paper is excellent and should be *accepted subject to minor revisions*.

**2   General comments**

1. **Experimental procedure and the texture of sea-ice:** There is insufficient discussion of how the procedure followed might have affected the texture of the sea ice, such that the images collected are not necessarily representative of natural sea ice. Several aspects of the description of the procedure raised questions about this matter. For example, L65 describes an equilibration over 1–3 days. L71 describes a loss of brine during storage. I would expect the storage period to result in change in texture or pore space geometry. Loss of brine will generally increase the solid fraction and reduce the permeability. L83–89 suggests that it might have been worthwhile varying the centrifuging procedure to demonstrate more clearly that results don't depend too sensitively on it.

2. **Porosity threshold** $\phi_c$**:** The experimental evidence provides very strong evidence that sea ice is permeable beneath the often-cited threshold of $\phi_c = 0.05$. However, the evidence in support of $\phi_c = 0.024$ is very much weaker. As an example, figure 6 and the text that discusses shows several samples beneath this critical threshold (to the left of the dashed red line). I'm not convinced it makes sense to extend the dashed red line outside of the data range, especially in panel b). I think the text should be altered to discuss this threshold more tentatively, perhaps arguing instead that any threshold must be smaller than about 0.024 (see also final point).

3. **Texture and the porosity threshold:** I think it remains an outstanding physical question whether a porosity threshold should be expected at all. I am more familiar with this discussion in the context of partially molten mantle rocks. For a

period in the 1990s, the dominant view was that there was a porosity threshold (e.g. Faul, 1997, JGR). But the general view today (partly as a result of experimental improvements) is that there is no such threshold. If the sample is at textural equilibrium, then the texture is controlled by the ratio of surface energy, often expressed as a dihedral angle. If this angle is beneath $60°$, the melt network remains connected to arbitrarily small porosities (Rudge, 2018, Proc. Roy. Soc., building on, e.g. von Bargen & Waff , 1986, JGR). The present study pushes the porosity threshold smaller than that suggested by previous studies, but perhaps rather close to the imaging threshold (see Table 3). Therefore, I would suggest that the conclusions/abstract should be more tentative. I would also expand section 4.5 to discuss the relationship further the relationship between texture and a threshold, building on the good discussion suggesting that the microstructure is controlled by morphological instabilities during ice growth rather than surface energy (in section 4.3). It would be good to see a bit more evidence for this claim and to consider whether the validity of this statement might evolve over time?

**3 Specific comments and technical corrections**

4. **L38–46:** This paragraph made me wonder whether it would be worth comparing this approach to laboratory permeameter measurements in future?

5. **L52:** 'not shown' is a typo?

6. **L58:** given the effort made to transport the samples rapidly, was any estimate made of the temperature change the samples might have experienced during the time

7. **L60:** consider noting that if samples were collected at a colder period, there would be a substantial *it situ* temperature gradient even across a 3.5 cm sample.

This procedure would need adapting for a colder collection period.

8. **L63:** I didn't have a clear sense as to why the samples were collected in an usually warm period? This should be explained at some point (perhaps it was not by design?) Does it limit the relevance of the results?

9. **L81:** I would make it clearer that the stated accuracy is an analytical or measurement accuracy. The sample treatment errors might be larger.

10. **L96:** What is $\phi_b$?

11. **L108 (footnote):** Do these approaches agree?

12. **L145:** I think this part should be explained more clearly. Ordinarily, the term 'hydrostatic pressure' refers to the part of the pressure that does not drive fluid flow.

13. **L155–156:** sentence structure could be clearer (perhaps missing a comma or split into two sentences).

14. **L187:** I think this makes sense, but perhaps explain the rationale for neglecting solid salts.

15. **L213:** typo/referencing issue.

16. **L226:** another occasion when the unusually temperature could be mentioned.

17. **Fig 4(b):** figure quality is very poor (hard to read).

18. **Fig 7 (and 9 and 10):** I found the legend confusing (e.g. what does (5) mean?)

19. **Fig 11:** Consider explaining what 'upward' and 'downward' mean in the figure caption (or refer readers to the main text).

20. **L298:** Perhaps clearer to say a factor of 100? Or 'two orders of magnitude'?

21. **L371:** Consider adding a reference for the 5% here.

22. **L414:** inconsistent italicisation of $D$ and $L$.

23. **L424:** missing space

24. **L505:** Consider adding a citation of Wettlaufer, J.S., Worster, M.G. & Huppert, H.E. 1997; The phase evolution of young sea ice; Geophysical Research Letters

25. **L577:** I'm not sure of the practicalities but it would be good to make the data available as soon as possible. https://wiki.pangaea.de/wiki/Data_submission seems to suggest that you could have 20 files each up to 100 MB here which might be suitable?

---

## Referee Comment (RC2) · Anonymous Referee #2 · 23 Dec 2020

This paper presents an interesting and detailed study of the connectivity properties of the porous brine microstructure of young natural sea ice via X-ray tomography of centrifuged samples, and the associated fluid transport properties of imaged reconstructions of the brine phase via numerical simulation. The results on the connectivity and fluid permeability at very low brine volume fractions and very small length scales are particularly significant, given the improved imaging resolution over previous studies of similar sea ice properties. This is a valuable study, a carefully written manuscript, and an important contribution to sea ice physics. However, I think the significance of this work as described in the abstract is somewhat misplaced, and implications for the so-called "rule of fives" that they draw from their results at very low brine volume

fractions and small scales are similarly off base and should be stated more carefully. Nevertheless, with a re-focus of some of the writing, results and conclusions, as well as careful consideration and addressing of the substantive specific issues raised below, I would recommend publication in The Cryosphere - again, after thoroughly revising the manuscript to take care of these concerns.

1. First, a general remark. Consider the two dimensional square bond lattice where bonds are open with probability p and closed with probability 1-p. In general, percolation thresholds are rigorously defined for infinite systems, with the threshold for the infinite square lattice of exactly 1/2. For a 10x10 sample of the lattice, there will be many realizations of the bond configurations where there exist paths of open bonds that connect one side to the other, even for p much less than 0.5. However, it can be proven for the infinite lattice that for any p<0.5, there does not exist a percolating (or infinite) cluster of open bonds, but that for p larger than or equal to 0.5 such a cluster does exist, which defines the threshold. Obtaining percolation thresholds or other critical points or even critical exponents from finite samples is a pervasive problem in statistical physics, and involves consideration of the correlation length and its relation to sample size, as discussed in detail for sea ice X-ray tomography in [Pringle et al., 2009]. One of my concerns about this paper is that there does not appear to be any consideration at all of the relationship between the one sample size they look at and their conclusions. Perhaps samples with vertical extent of 8 cm (if they could have scanned those) might typically require a brine volume fraction of 3.5% for there to be connections from top to bottom which include the micro-scale features that have been resolved with their instrument and analysis. Figure 3 in [Pringle et al., 2009] shows a transition around brine volume fraction of about 5% in the behavior of the fractional connectivity (fraction of brine voxels on one face connected to the opposite face, regardless of path characteristics) for sea ice single crystals, and its dependence on sample size, and a corresponding divergence of the correlation length as 5% is approached from below. Do you have data below 2.4% that shows a similar transition or correlation length divergence as you approach the threshold from below, which would
support the notion of a small scale threshold at 2.4%? Or if one could accurately image even smaller features, would one find an even smaller threshold, or a series of smaller thresholds?

2. For binary lattice percolation models, such as the 2D square lattice with bonds open or closed, given a finite sample and a bond configuration, there either is or is not a path of open bonds connecting one side of the sample to the other. However, if the bonds are pipes with arbitrarily small radii, that is, the radii are chosen from a probability distribution with support down to 0, then the question of whether a configuration or cluster percolates or not is now determined by how "thick" one requires the spanning pathways in the cluster to be. In other words, given a cut-off radius, one can then ask if connected clusters of pipes whose radii exceed the cut-off percolate or not. If they do, then fluid flowing through a percolating cluster of large enough pipes will generally be forced to travel through some of the smallest pipes whose radii are near or at the cut-off. Moreover, these "bottlenecks" or throats determine the leading order behavior of the fluid permeability, or the effective electrical conductivity if the bonds are conductors. There are rigorous theorems (and analogous techniques in theoretical solid state physics) to this effect that form the basis of critical path analysis [Golden and Kozlov, in Homogenization: Serguei Kozlov Memorial Volume, V. Berdichevsky et al. (Eds.), 1999; Golden, in Homogenization and Porous Media, U. Hornung (Ed.), 1997; Golden et al., GRL, 2007; Ambegaokar, Halperin and Langer, Phys. Rev. B, 1971]. In the context of sea ice, which is of course a continuum material, the percolation characteristics of the porous brine microstructure can be thought of in terms of the pipe network described above. One way of putting a principal result of this paper is that in sea ice samples of vertical dimension 2 cm to 3 cm with brine volume fractions exceeding 2.4%, there are fluid pathways through the brine phase connecting the top to the bottom whose minimal "diameter" exceeds 0.07 mm (or in terms of the pipe network, configurations of pipes whose diameters exceed a cut-off of 0.07 mm span the sample vertically, or percolate).

Now, let us discuss how this result is related to the so-called "rule of fives" and the generally accepted value of 5% brine volume fraction for the "percolation threshold" of sea ice. The concise statement of this "rule of thumb" in the first paragraphs of the papers [Golden et al., Science, 1998; Golden et al., GRL, 2007, Pringle et al., JGR, 2009] is that columnar sea ice is "effectively impermeable" to bulk fluid flow for brine volume fractions below about 5%. This is not stated as a mathematical theorem, and there is an understanding that by the very nature of percolation theory for finite samples, and the complex multiscale structure of the brine phase, one would expect the possibility of some fluid flow over relatively small scales through relatively small pore spaces, even for brine volume fractions below 5%. (Figure 3b in [Golden et al., 2007] shows that the fractional connectivity for samples of 8 mm in vertical extent remains non-zero down to below 4% brine volume fraction.) However, I have personally made hundreds of in situ measurements of the vertical fluid permeability of sea ice in the Arctic and Antarctic, by removing partial cores and then measuring the rate at which water fills the hole through the ice at the bottom of the hole by various techniques. Even with the uncertainties in this "sack hole" method, I can unequivocally state that if the sea ice at the bottom of the hole is columnar and has brine volume fraction below about 4% or 5% (and horizontal flow is blocked with "packers"), there will most likely be very little or no measurable fluid in the hole even after a few hours. As the brine volume fraction of columnar sea ice decreases from high values associated with quite permeable ice, there is a noticeable, clear transition to bulk fluid flow over the scale of tens of centimeters relevant to the experiment, being essentially shut down, or the ice becoming effectively impermeable, once the brine volume fraction gets below about 5%. Roughly speaking, permeability values then generally lie below about $10^{-12}$.

The spirit in which this rule was developed was in terms of whether or not the brine microstructure could enable various geophysical processes such as surface flooding and subsequent snow-ice formation, melt pond drainage, and changes in salinity. For example, suppose we consider upward percolation of sea water and brine due to snow loading of the ice surface, and the subsequent freezing of the flooded surface snow

layer. If the upper layer of sea ice through which fluid must pass to reach the surface, say, has permeability around 10ˆ{-13} with brine volume fraction just below the threshold, we may wind up with very little water on the surface and essentially no new snow-ice production. On the other hand, if the permeability of this restrictive layer is around 10ˆ{-11} or larger with brine volume fraction a bit above the threshold, then after several hours there may be a few centimeters of water on the surface which could produce a significant amount of snow-ice that affects ice mass-balance accounting.

From the point of view of the pipe network, the 5% threshold for bulk flow in practice means that in sea ice samples of vertical dimension on the order of, say, 20 cm to 50 cm with brine volume fractions above about 5%, there are fluid pathways through the brine phase connecting the top to the bottom whose minimal "diameter" exceeds a certain cut-off value, which is much larger than 0.07 mm. As reported in [Weeks and Ackley, 1982], S. Martin and co-workers over many studies found that most vertically oriented "channels" through which the bulk of fluid is transported through sea ice over tens of centimeters have diameters that range from about 1 mm to 1 cm, with some channels much larger. In Equation (4) of [Golden et al., GRL, 2007] the cut-off, bottleneck, or minimal diameter was then chosen to be 1 mm, which leads via critical path analysis to the prediction of the scaling factor in front. This percolation formula in Equation (4) with a bulk transport threshold of 5% then agrees very closely with in situ data for brine volume fractions above the threshold. By way of comparison with the scales considered in the current paper, the amount of fluid that flows per unit time through a circular pipe of diameter 1 mm is 10ˆ{4} times the amount that flows per unit time through a pipe of diameter 0.1 mm, which is just a bit larger than the critical diameter of 0.07 mm in this paper.

Thus, it is not really appropriate to state without careful explanation and qualification that the 2.4% value found here is considerably lower than the 5% threshold which has been widely used for bulk flow over larger scales. What is being referred to is quite different for these two situations, in the setting of a multiscale porous medium like sea

ice as described above, with the rule of fives and the 5% threshold describing fluid transport behavior on significantly larger sample and pore size scales than what is considered in this paper. (In fact, the vertical sample size of 5 mm for the permeability simulations calls into question the applicability of this work beyond the smallest of scales. See Figure 3 in [Pringle et al., 2009] on the dependence of the correlation length with brine volume fraction.) The results described in [Golden et al. 2007, Pringle et al., 2009] with a vertical threshold of 4.6% (and higher thresholds in the horizontal directions) were a first step in imaging the connectivity of the brine phase and building toward larger scales, with the analysis of the most basic building blocks - sea ice single crystals. As stated in [Golden et al. 2007], "These images provide insight into and constraints for more detailed modeling of micro-scale inclusion connectivity. A similar analysis of large-scale pore networks remains challenging, but is inherently reflected in the in situ permeability data." Indeed, extending such analyses to the scales relevant for the rule of fives remains a challenge today.

3. One major significance of the paper, in my opinion, is that they have explored a new level of fine scale structure and conducted a high resolution analysis of the habitats of microbial life in sea ice, and also carefully computed a key property of sea ice which is critical for local nutrient fluxes, namely, fluid permeability, on scales which may be particularly relevant for small scale biological processes.

4. I am concerned that while centrifuging may leave major inclusion structures intact, if this process modifies the brine microstructure by opening up new pathways that then appear to be connections in the X-ray tomographic images, it will be at these finest scales that are the focus of this paper. I think there should be some discussion of how the centrifuging process may or may not affect the 2.4% value thus obtained.

5. The results in [Perovich and Gow, JGR, 1996] should be referenced and briefly discussed in light of the results in this paper.

6. A few sentences, or synopsis, about the scales of features the authors can resolve

and how their resolution compares to previous works would be welcome, or if this were made a little clearer with a sentence like: "The inclusions we see that form connected fluid pathways at lower brine volume fractions than have been observed before have the following characteristic ranges of dimensions, etc." The paragraph around line 135 needs to be expanded and clarified.

7. In line 375, it is stated, "So far sea ice permeability has been studied in terms of isotropic percolation (Petrich et al., 2006; Golden et al., 2007; Pringle et al., 2009)". However, in [Pringle et al., JGR, 2009] the finding of different values of the percolation threshold in three perpendicular directions for a sea ice single crystal is certainly anisotropic percolation, and should be noted as such.

8. The idea of analyzing the effective centrifuged brine volume fraction compared to the total is excellent. However, by a certain point there seem to be so many different types of porosities running around that it is difficult to keep them straight. Perhaps a synopsis and overall explanation would be helpful, as well as clearer definitions with diagrams of all the parameters in Table 2.

9. Line 23 - should refer to [Polashenski et al., JGR, 2018] which deals explicitly with the issue of how initially permeable sea ice supports melt ponds.

10. Line 92 - missing a "c" in subscript of phi.

11. The exponent in Equation (8) is 2.6, which I assume is a best fit. The corresponding exponent in Equation (4) in [Golden et al., GRL, 2007] is 2. This is a theoretical prediction, based on an argument that even though sea ice is a continuum material that could exhibit so-called non-universal behavior different from lattices (with its exponent larger than 2, see [Golden, PRL, 1990] for rigorous results on lattices), it exhibited universal behavior due to the lognormal distribution of brine inclusion sizes. It is interesting to speculate if the exponent of 2.6 in this paper is a demonstration of non-universal behavior at these fine scales. Perhaps a sentence could be added addressing this?

12. Line 15 - I believe the authors meant to say "Sea ice is a porous medium that covers, on average, about 12 percent of the earth's oceans." (Or about 7 percent of earth's surface).

---

## Author Comment (AC1) · 23 Mar 2021

**Answer to Reviewer 1 of tc-2020-288**

We like to thank the reviewer for this detailed review that helped us considerably to improve the paper and clarify open issues. Below we repeat the *reviewers comments in italic font* followed by our answers, and **additions and changes to the manuscript in red font**.

**1 Summary**

*This paper presents a new experimental characterization of the pore space and permeability of natural sea ice. The techniques are advanced and novel for natural sea ice. These measurements have a wide importance to those studying the evolution of sea ice, since many processes are very sensitive to permeability. The experimental technique is well described and careful. The results are presented clearly and provide strong evidence in contrast to widely-cited previous studies showing a percolation threshold at 5 % porosity. The limitations of the study are discussed very well, although there are three significant areas in which I think limitations need more discussion. The writing quality is excellent. Overall, I think the paper is excellent and should be accepted subject to minor revisions.*

**Answer:**
Thank you for this motivating evaluation. We agree with most aspects to be revised, and answer the reviewers comments below. In advance we give the modified abstract:

**Abstract.** The hydraulic permeability of sea ice is an important property that influences the role of sea ice in the environment in many ways. As it is difficult to measure, so far not many observations exist and the quality of deduced empirical relationships between porosity and permeability is unknown. The present work presents a study of the permeability of young sea ice based on the combination of **brine extraction in a centrifuge,** X-ray micro-tomographic imaging and direct numerical simulations. The approach is new for sea ice. It allows to relate the permeability and percolation properties explicitly to characteristic properties of the sea ice pore space, in particular to pore size and connectivity metrics. For the young sea ice from the present field study we obtain a brine volume of **2 to 3%** as threshold for the vertical permeability (transition to impermeable sea ice). We are able to relate this transition to the necking of brine pores at a critical pore throat diameter of $\approx 0.07$ mm, being consistent with some limited pore analysis from earlier studies. **Our optimal estimate of critical brine porosity is half the value of 5 %** proposed in earlier work and frequently adopted in sea ice model studies and applications. **From a discussion of our results with respect to earlier studies we conclude that the present threshold is more significant, in particular through the combination of 3D image analysis and centrifuge experiments. We also find some evidence that the sea ice pore space should be described by *directed* rather than *isotropic* percolation. Our revised porosity threshold is valid for the permeability of young columnar sea ice dominated by primary pores. For older sea ice containing wider secondary brine channels, for granular sea ice, as well as for the full thickness bulk permeability, other thresholds may apply**.

**2 General comments**

**2.1 Experimental procedure and the texture of sea-ice:**

*There is insufficient discussion of how the procedure followed might have affected the texture of the sea ice, such that the images collected are not necessarily representative of natural sea ice. Several aspects of the description of the procedure raised questions about this matter. For example, L65 describes an equilibration over 1-3 days. L71 describes a loss of brine during storage. I would expect the storage period to result in change in texture or pore space geometry. Loss of brine will generally increase the solid fraction and reduce the permeability. L83-89 suggests that it might have been worthwhile varying the centrifuging procedure to demonstrate more clearly that results don't depend too sensitively on it.*

**Answer:**

We have written a more detailed description of the procedure of sampling, transport, centrifugation and storage. First, we expand the methods section by an additional paragraph:

L137 **2.5. Sampling, transport, storage and textural changes: Special care was taken to minimise undesired temperature changes and variability prior to centrifugation and imaging. The cut samples of the relatively isothermal sea ice were transported in an isopleth box (inside a larger insulated aluminium box) to the laboratory. Transport and sorting into small temperature controlled freezers happened within half an hour. As each sub sample was packed in a conical plastic cup, temperature changes are, due to the large effective specific heat capacity, considered negligible. The box temperature was logged by a temperature logger, as well as temperatures were directly measured on samples, being within 0.2 K of in sit values. The next step, cooling of sub samples in the laboratory, took place within these freezers set to lower than in sit temperatures. With samples within the plastic cups cooling rates (with most heat loss due to internal freezing) were moderate and in the range 1-5 K per day, comparable to natural cooling rates. An important aspect of the approach was also that samples were only cooled, not warmed. This avoids the known hysteresis, that brine expelled during cooling is not reintroduced into a sample upon warming.**

**Though we have no strict proof for this, we believe that microstructure changes during 1 to 2 days of close to isothermal storage are minor (this is based on unpublished work of repeated scanning). More relevant could be effects due to freezing and redistribution of brine. First, one could expect that simultaneous cooling of sub samples from all sides may redistribute brine in a way that differs from mostly vertical heat loss of ice in the field. We do not find brine accumulation in the center of samples, indicating that also the multi-directional sample cooling redistributes brine along the predominantly vertically oriented pores. Brine could be redistributed vertically in some non-uniform way within a 3 cm thick sub sample, and implications will be considered in the discussion. Second, we treat our sample isothermally, which is justified as the in sit temperature profile suggests a difference of 0.1 K along the vertical direction. Third, sample storage after centrifugation at**

low temperature (-80 ℃) has likely led to almost complete precipitation of all residual brine. During XRT imaging these salt crystals have dissolved again. As the microstructure of these pores will very likely differ from field values, we do not analyse it here. We regard it as unlikely, that this hysteresis of disconnected pores has affected the networks of connected pores.

We finally note that the small in sit temperature range made this study logistically easier as if the ice had been sampled during a cold period.

Second, loss of brine only would increase the solid fraction if replaced by freezing sea-water. In our case, loss of brine simply results in air-filled open channels, that are not distinguished from channels emptied during centrifuging. They are identified as open, in sit filled with brine, and contribute to permeability in our simulations. However, we add a note about the relevance of drained brine in the description of the centrifuge procedure: L103 The centrifuged brine mass on which the effective porosity is based also includes brine that has leaked from the sample during storage, prior to centrifuging. In our study this pre-drained brine volume was not negligible and contributed on average 28% of the total (leaked and centrifuged) brine volume. On the one hand this value may be an overestimate, as it could include small ice particles that fell into the cup during sampling. On the other hand, there is very likely some brine lost during coring and cutting, which will underestimate the centrifuge-based effective porosity. Both effects imply a difference between CT-based and centrifuge based estimated of effective porosity that we cannot resolve with our data.

Third, we clarify the information about the centrifuging procedure.
L89 The centrifuge parameters depend on centrifuge type and were carefully chosen on the basis of several tests. (i) Ice samples were centrifuged with temperature loggers to determine temperature stability. Slight warming of the centrifuge was observed, leading us to the choice of an in initial centrifuge temperature 1K below the ice in-situ temperature. A similar value was chosen by Weissenberger et al. (1992) similar centrifuge times. (ii) Varying the centrifuging time from 10 to 20 minutes showed that more than 95% of brine where extracted during the first ten minutes, and we selected 15 minutes. (iii) Freitag (1999) noted that incomplete centrifugation of brine might lead to brine remnants which, after cooling and freezing, might block pores and decrease the permeability. We have indeed found such a result in an earlier study with centrifuge acceleration of 15 g (Buettner, 2011) and thus tested the effect of relative centrifuge acceleration for three ice cores at 10, 25 and 40 g. The result was on average 20% less centrifuged brine at 10 g, but only a slight non-significant 5% difference between 25 and 40 g. We thus are confident that 40 g is a proper choice for extracting the connected brine.

**2.2 Porosity threshold $\phi_c$:**

*The experimental evidence provides very strong evidence that sea ice is permeable beneath the often-cited threshold of $\phi_c = 0.05$. However, the evidence in support of $\phi_c = 0.024$ is very much weaker. As an example, figure 6 and the text that discusses shows several samples beneath this critical threshold (to the left of the dashed red line). I'm not convinced*

*it makes sense to extend the dashed red line outside of the data range, especially in panel b). I think the text should be altered to discuss this threshold more tentatively, perhaps arguing instead that any threshold must be smaller than about 0.024 (see also final point).*

**Answer:**
We agree that the optimal threshold porosity $\phi_c = 0.024$ cannot be deduced from the CT measurements alone as these data are scattered, and the CT samples are $1/50$ in volume of the centrifuged samples (L266-269). However, the evidence based on the larger centrifuged samples is much stronger. We base our confidence on the the power law fit of the centrifuged porosity in Figure 4a (critical exponent $\beta = 0.832$) and its consistency with the theoretical critical exponent from direct percolation. We are stating more carefully that, if the sea ice pore space can be characterised by directed percolation (with a theoretical critical exponent $\beta \sim 0.82$) then the centrifuge experiments are consistent with a threshold in the range $0.020 < \phi_c = 0.029$. We have now determined these wider confidence bounds as described in 3.2.1:

L245 **For the critical $\phi_c$ we obtain confidence bounds by using $0.803 < \beta < 0.861$ and $\phi_c = 0.0240$ as input to a power law regression, which in turn resulted in 95% bound range of $0.20 < \phi_c < 0.29$.**

We add the following to beginning of 4.5:

L513 **The present analysis has enabled us to deduce a porosity threshold of $2.0 < \phi_c < 2.9\%$. This optimal threshold cannot be deduced from the CT measurements alone as these data are scattered, and the CT samples are $1/50$ in volume of the centrifuged samples. However, the evidence based on the larger centrifuged samples is much stronger. Our confidence is related to the power law fit of the centrifuged porosity in Figure 4 (critical exponent $\beta = 0.83 \pm 0.03$) and its consistency with the theoretical critical exponent from directed percolation (critical exponent $\beta = 0.82$). Based on this agreement we can state that, if our hypothesis is correct, that the pore space evolution of sea ice follows the behaviour of *directed percolation*, then this implies a threshold porosity for percolation in the range 2 to 3 %. The CT-based microstructure analysis supports these results, and further can be interpreted in the way, that the threshold is related to the necking or close-off of pores at a critical diameter of 0.07 mm. Our deduced porosity threshold is just half of the value of $\phi_c = 5\%$ once proposed by Golden et al. (1998), that since then has been confirmed in other studies and become the mostly accepted threshold for sea ice permeability and desalination (Weeks, 2010). In the following we discuss these studies in the context of our results.**

We have modified the abstract to make clearer, that the threshold is based on the combination of centrifuge experiment and 3D imaging. We finally note that we recently submitted a manuscript on X-ray tomography of another sea ice core set (M. L. Salomon, S. Maus and C. Petrich, 'An Investigation of the Microstructure Evolution of Young Sea Ice from a Svalbard Fjord', submitted to the Journal of Glaciology, March 2021). The latter focuses on higher porosities, with few data points near the threshold, but the latter are consistent with the present study.

**2.3 Texture and porosity threshold:**

*I think it remains an outstanding physical question whether a porosity threshold should*

*be expected at all. I am more familiar with this discussion in the context of partially molten mantle rocks. For a period in the 1990s, the dominant view was that there was a porosity threshold (e.g. Faul, 1997, JGR). But the general view today (partly as a result of experimental improvements) is that there is no such threshold. If the sample is at textural equilibrium, then the texture is controlled by the ratio of surface energy, often expressed as a dihedral angle. If this angle is beneath 60°, the melt network remains connected to arbitrarily small porosities (Rudge, 2018, Proc. Roy. Soc., building on, e.g. von Bargen and Waff, 1986, JGR). The present study pushes the porosity threshold smaller than that suggested by previous studies, but perhaps rather close to the imaging threshold (see Table 3). Therefore, I would suggest that the conclusions/abstract should be more tentative. I would also expand section 4.5 to discuss the relationship further the relationship between texture and a threshold, building on the good discussion suggesting that the microstructure is controlled by morphological instabilities during ice growth rather than surface energy (in section 4.3). It would be good to see a bit more evidence for this claim and to consider whether the validity of this statement might evolve over time? .*

**Answer:**

We agree that the conclusions/abstract in terms of the porosity threshold should be more tentative. We have rewritten and extended 4.5. for the porosity threshold and added a section 4.6., where we clarify the need to distinguish between granular and columnar ice texture, young and old ice and local and full-thickness permeability, and discuss (requested by the second reviewer) scale effects. We point out that our results only apply to columnar young sea ice, and add figure 10 and its description in 3.4 to make the overall pore size distribution of our ice more clear. However, many of the studies that had led to the conjecture of $\phi_c$ 5% were studies of young ice, and our general conclusion to revise this threshold remain the same.

Description of figure 10 in subsection 3.4:

[revised manuscript text omitted]

With regard to morphological stability and future needs and options we add at the end of 4.3.: L452 Due to these factors it seems more likely that morphological freezing instabilities in supercooled brine layers play a role for the necking, in the similar way as they do for the plate or brine layer spacing at the ice-water interface (Wettlaufer, 1992; Maus, 2020). A concise physical explanation for the necking phenomenon is lacking so far. Progress could be made by direct observations of the 3D pore space evolution by X-ray tomography, building on 2D visual observations of pore necking described by Niedrauer and Martin (1979) for a thin growth cell. Such an approach may be feasible through fast time-lapse synchrotron-based X-ray tomography for low contrast materials (Beckmann et al., 2007; Buffiere et al., 2010, e.g.,). Also conventional laboratory-based XRT with higher spatio-temporal resolution may provide new insight into the details of necking and pore instabilities.

**2.4 Specific comments and technical corrections**

*4. L38-46: This paragraph made me wonder whether it would be worth comparing this approach to laboratory permeameter measurements in future?*
We are adding the following suggestion for future work in the conclusions:
L574 The present work presents new insight into the sea ice pore space evolution of young sea ice and theoretical interpretation of the latter. It demonstrates the large potential of 3D X-ray micro-tomographic imaging to make progress in our fundamental understanding of sea ice properties. For future work we suggest several directions to make further progress. First of all, the result for permeability-porosity relations and thresholds, are not directly transferable to the thicker, older and warmer summer ice. The latter often contains coarser secondary brine channels that are lacking in young ice and

that are relevant for processes melt pond percolation and melt pond albedo feedbacks (Freitag and Eicken, 2003; Polashenski et al., 2017). This reflects one of the challenges in sea ice physics, which is to improve our understanding how the sea ice pore space, as well as permeability and other physical properties evolve over time. To make progress more 3D CT data of sea ice and its pore space evolution over time are needed. Second, while the present study may be seen as a starting point to a concise understanding and modelling of this evolution, it should be verified with higher spatial resolution to clarify any resolution limit with respect to necking and porosity thresholds. Third, there is a need for comparing granular and columnar ice, as the granular surface layer will be important for the through-flow permeability. And last but not least, due to the lack in experimental data, carefully controlled laboratory experiments in the lines of Ono and Kasai (1985) would be of high value. Combing such experiments with repeated CT imaging to monitor flow-induced microstructure changes, could provide valuable insight about the evolution of the sea ice pore space and its permeability. A useful concept with centrifuged sea ice would be measurements of kerosine permeability in a permeameter (Saito and Ono, 1978; Saeki et al., 1986), allowing validation of permeability computed from CT imagery and the question if there are sub-resolution pathways.

Also a slight change to the conclusions was made for clarifying our technique:

L571 **The connected and open air fractions are deduced by XRT image analysis, and the connected air is associated with connected brine. The approach also considerably increases the image quality, as XRT imaging of brine networks at high temperatures suffers from contrast limitation between ice and brine (Pringle et al., 2009; Bartels-Rausch et al., 2014).**

*5. L52: 'not shown' is a typo?*
Ok, removed.

*6. L58: given the effort made to transport the samples rapidly, was any estimate made of the temperature change the samples might have experienced during the time*
See next comment.

*7. L60: consider noting that if samples were collected at a colder period, there would be a substantial it situ temperature gradient even across a 3.5 cm sample. This procedure would need adapting for a colder collection period.*
We added, in addition to the novel subsection 2.5., the following:
L59 **For the given field conditions the temperature change that samples may have experienced during this transport might be a few tens of a Kelvin (see below in section 2.5). We note that less isothermal ice would have required a more advanced temperature control of the different levels in the ice.**

*8. L63: I didn't have a clear sense as to why the samples were collected in an usually warm period? This should be explained at some point (perhaps it was not by design?) Does it limit the relevance of the results?*
We add:

L54 **The insulation through the snow cover resulted, in spite of air temperatures varying by 7 K during the sampling period, in only minor ice temperature changes over 5 days, and a temperature range of less than 1 K over 35 cm thickness. While originally sampling of ice at different temperatures was planned, the stable temperature turned out as an advantage for temperature control, and allowed to harvest and analyse ice cores of very similar salinity and structure, and rather to perform a controlled cooling sequence in the laboratory.**

*9. L81: I would make it clearer that the stated accuracy is an analytical or measurement accuracy. The sample treatment errors might be larger.*
We add the word L81 **measurement** to accuracy:

*10. L96: What is $\phi_b$?*
Should be $\phi_b$, changed.

*11. L108 (footnote): Do these approaches agree?*
We add:
L108 **We tested also this approach and only found relative differences of a few percent.**

*12. L145: I think this part should be explained more clearly. Ordinarily, the term 'hydrostatic pressure' refers to the part of the pressure that does not drive fluid flow.*
We remove 'hydrostatic' and keep the formulation, as we have referred to porous media textbooks

*13. L155-156: sentence structure could be clearer (perhaps missing a comma or split into two sentences).*

We reformulate:
L155 **In ice samples with a lower permeability than that value, one can expect incomplete removal of brine. Upon cooling this brine will partially freeze and may render the sample impermeable.**

*14. L187: I think this makes sense, but perhaps explain the rationale for neglecting solid salts.*
We add:
L187 **Including solid salts in the calculations would decrease brine volume fractions at the lower end of our porosity range by 0.1-0.2 % (Cox and Weeks, 1983, see), and have little effect on our results.**

*15. L213: typo/referencing issue.*
Corrected.

*16. L226: another occasion when the unusually temperature could be mentioned.*
We add:
L227 **Note that, due to a 10 cm snow cover, the ice temperature from the other two sampling dates, two days earlier and later, was very similar**.

*17. Fig 4(b): figure quality is very poor (hard to read).*

Lines and font have been adjusted. We added also in the caption:

Caption Fig. 7 **Optimum exponent $\beta$ in dependence on porosity threshold $\phi_c$ and the $R^2$ of double-logarithmic least square fits of $\phi_{cen}$ versus $(\phi - \phi_c)^{\beta}$. The point of maximum correlation is shown as a star.**

*18. Fig 7 (and 9 and 10): I found the legend confusing (e.g. what does (5) mean?)*
We checked that in all Figures the shown curves and legends are properly described. The following should explain the legend better:

Caption Fig. 7 **Two log-log fits are drawn and specified in the legend, corresponding to power laws of the form $K = a(\phi)^b$ (green curve), and $K = a(\phi - \phi_c)^b$ as red curve. The numbers in brackets are the uncertainties in the last decimal of the log-log least square fit.**

*19. Fig 11: Consider explaining what 'upward' and 'downward' mean in the figure caption (or refer readers to the main text).*
We extend the figure caption to:

Caption Fig. 11 **Relationship of simulated vertical permeability $K$ and $(\phi - \phi_c)$ as shown in Figure 7 as red dashed curve, here compared to two earlier investigations. As the measurements from Freitag (1999) are only available in terms of $\phi_{cent}$, we plot the least square fit from Freitag (cyan curve, numbers given in the legend). The data points from Ono and Kasai (1985) represent downward and upward permeation of brine through the ice - see text for more information.**

*20. L298: Perhaps clearer to say a factor of 100? Or 'two orders of magnitude'?*
We correct to 'two orders of magnitude'.

*21. L371: Consider adding a reference for the 5 % here.*
References added (the same ones as given in other parts of the paper).

*22. L414: inconsistent italicisation of D and L.*
Corrected.

*23. L424: missing space*
Corrected.

*24. L505: Consider adding a citation of Wettlaufer, J.S., Worster, M.G. and Huppert, H.E. 1997; The phase evolution of young sea ice; Geophysical Research Letters*
These authors indeed discussed a permeability function and we add this reference.

*25. L577: I'm not sure of the practicalities but it would be good to make the data available as soon as possible. https://wiki.pangaea.de/wiki/Data-submission seems to suggest that you could have 20 files each up to 100 MB here which might be suitable?*
The raw data files have a total size of over 670 GigaBytes. However, we are going to publish the cropped files (170 Gigabytes), on which the present analysis is based, in the open NIRD research data archive. This is under processing and the DOI will be added in the revised manuscript.

[Figure]

Figure 1: L513 **New Figure 10** Pore size distributions based on XRT imaging of 4 young ice coresles. a) Fraction of open brine pores in 18 $\mu$m wide size bins for the two warmest (red) and two coldest (blue) cores. The corresponding cumulative fractions are also shown, with y-axis on the right hand side; b) same as a) but for the porosimetry/ fraction of pores throats.

Saeki, H., Takeuchi, T., Sakai, M., Suenaga, E., 1986. Experimental study on the permeability coefficient of sea ice. In: T.K.S. Murthy, J. C., C.A.Brebia (Eds.), Ice Technology, Proceedings: 1st International Conference. Springer-Verlag, New York, pp. 237–246.

Saito, T., Ono, N., 1978. Percolation of sea ice. I: Measurement of kerosine permeability of nacl ice. Low Temp. Sci. A37, 55–62.

Stauffer, D., Aharony, A., 1992. Introduction to Percolation Theory, 2nd Edition. Taylor & Francis.

Weeks, W. F., 2010. On Sea Ice. University of Alaska Press.

Weissenberger, J., Dieckmann, G., Gradinger, R., Spindler, M., 1992. Sea ice: A cast technique to examine and analyze brine pockets and channel structure. Limnol. Ocean. 37 (1), 179–183.

Wettlaufer, J. S., 1992. Directional solidification of salt water: deep and shallow cells. Europhysics Letters 19 (4), 337–342.

[Figure]

Figure 2: **New Figure 13** Comparison of simulated vertical permeability $K$ and $(\phi - \phi_c)$ for columnar samples (as shown in Figure **??** and here as open circles) to two other simulation results. The solid circles are harmonic means of all subsamples (normally four to five) from each basic sample cut in the field (and centrifuged). The red stars are permeability results for near-surface samples (up to 5 cm from the surface) that were classified as granular ice and thus excluded from the basic analysis.

---

## Author Comment (AC2) · 23 Mar 2021

**Answer to Reviewer 2 of tc-2020-288**

We like to thank the reviewer for this detailed review that helped us considerably to improve the paper and clarify open issues. Below we repeat the *reviewers comments in italic font* followed by our answers, and **additions and changes to the manuscript in red font**.

**1 Summary**

*This paper presents an interesting and detailed study of the connectivity properties of the porous brine microstructure of young natural sea ice via X-ray tomography of centrifuged samples, and the associated fluid transport properties of imaged reconstructions of the brine phase via numerical simulation. The results on the connectivity and fluid permeability at very low brine volume fractions and very small length scales are particularly significant, given the improved imaging resolution over previous studies of similar sea ice properties. This is a valuable study, a carefully written manuscript, and an important contribution to sea ice physics. However, I think the significance of this work as described in the abstract is somewhat misplaced, and implications for the so-called 'rule of fives' that they draw from their results at very low brine volume fractions and small scales are similarly off base and should be stated more carefully. Nevertheless, with a re-focus of some of the writing, results and conclusions, as well as careful consideration and addressing of the substantive specific issues raised below, I would recommend publication in The Cryosphere - again, after thoroughly revising the manuscript to take care of these concerns.*

**Answer:**

We agree with the reviewer in most but not all aspects. We like to say first that our contribution is not meant as a general critics of the 'rule of the fives' and the great idea of linking this rule of thumb to percolation-based modelling of sea ice. In particular the first author of this manuscript also likes to point out that he has learned a lot from the publications on sea ice permeability and the ideas on percolation proposed by K. Golden and his colleagues (Golden et al., 1998, 2007; Petrich et al., 2006), and has been working for many years to now be able to extend and refine these concepts. First now this is possible through larger and higher resolution 3D imagery of the sea ice pore space, higher performance computing as well as theoretical developments that account for the directional character of sea ice percolation. We are now highlighting this in the beginning of the conclusions:

L551 **Most previous investigations on the permeability of sea ice have been facing challenges related to difficulties to observe this property in situ, and to the reactive nature of sea ice during transport, storage and experiments. The 'rule of the fives' proposed by Golden et al. (1998) has provided, together with ideas to model sea ice on the basis of percolation theory (Golden et al., 1998, 2007; Petrich et al., 2006; Pringle et al., 2009), an attractive and reasonable rule of thumb to understand this important sea ice property better. However, much of these results have been based on indirect observations (like surface flooding and desalination) and a stringent validation has been lacking for sea ice and its different micro-structures. First now it has been possible for us, through larger and higher resolution 3D imagery and higher perfor-**

**mance computing, to study the pore scale details of sea ice and perform direct numerical simulations of its permeability.**

We agree that a more careful formulation is needed for which ice properties we propose a revised percolation threshold, and will clarify for which ice type, age and scales the present results are valid. However, we do not agree that our proposed *implications for the so-called 'rule of fives' that they draw from their results at very low brine volume fractions and small scales are similarly off base*. We note that the 'rule of the fives' has once been argued on the basis of a conjecture about percolation on small scales (compressed powder model) and young sea ice, see figure 1 in ref. Golden et al. (1998) and their discussion. Also a later CT-based analysis that also concluded with the 5% threshold was based on young laboratory grown ice and small scales (Pringle et al., 2009). Our study refers to 'young sea ice' as stated in the title and it is this ice type for which we think the 'rule of fives' needs to be revised. This conclusion is not only based on our own data analysis, but also on a critical discussion of published data. E.g., the data from ref. Ono and Kasai (1985) that have been used to underline the 5% porosity threshold and 'rule of the fives', see figure 2B in ref. Golden et al. (1998), are unlikely from a porosity regime to allow such a conclusion (see our Fig. 11). Moreover, the results from Pringle et al. (2009) should, in view of our two times larger spatial resolution, be viewed with caution, as they may not have resolved many of the small pores that were permeable above our threshold.

Hence, the 'rule of the fives' was first based on considerations on microstructure and small scales (Golden et al., 1998) and has later been proposed for larger scales and thicker ice (Golden et al., 2007). Our analysis is for young ice and small scales, yet we think that the 'rule of the fives' should also be carefully revised for other ice types and age as well, as the evidence published so far is not too strong. E.g. the permeability data presented by Golden et al. (2007), their Figure 1 and 4, indicate that thicker ice is permeable below a porosity of 5%. In advance to our answers below we give the modified abstract:

**Abstract.** The hydraulic permeability of sea ice is an important property that influences the role of sea ice in the environment in many ways. As it is difficult to measure, so far not many observations exist and the quality of deduced empirical relationships between porosity and permeability is unknown. The present work presents a study of the permeability of young sea ice based on the combination of **brine extraction in a centrifuge,** X-ray micro-tomographic imaging and direct numerical simulations. The approach is new for sea ice. It allows to relate the permeability and percolation properties explicitly to characteristic properties of the sea ice pore space, in particular to pore size and connectivity metrics. For the young sea ice from the present field study we obtain a brine volume of **2 to 3%** as threshold for the vertical permeability (transition to impermeable sea ice). We are able to relate this transition to the necking of brine pores at a critical pore throat diameter of $\approx 0.07$ mm, being consistent with some limited pore analysis from earlier studies. **Our optimal estimate of critical brine porosity is half the value of 5 %** proposed in earlier work and frequently adopted in sea ice model studies and applications. **From a discussion of our results with respect to earlier studies we conclude that the present threshold is more significant, in particular through the combination of 3D image analysis and centrifuge experiments. We also find some evidence that the sea ice pore space should be described by *directed* rather than *isotropic* percolation. Our revised porosity threshold is valid for the permeability of young columnar sea ice dominated by primary pores. For**

**older sea ice containing wider secondary brine channels, for granular sea ice, as well as for the full thickness bulk permeability, other thresholds may apply**.

*1. First, a general remark. Consider the two dimensional square bond lattice where bonds are open with probability p and closed with probability 1-p. In general, percolation thresholds are rigorously defined for infinite systems, with the threshold for the infinite square lattice of exactly 1/2. For a 10x10 sample of the lattice, there will be many realizations of the bond configurations where there exist paths of open bonds that connect one side to the other, even for p much less than 0.5. However, it can be proven for the infinite lattice that for any p<0.5, there does not exist a percolating (or infinite) cluster of open bonds, but that for p larger than or equal to 0.5 such a cluster does exist, which defines the threshold. Obtaining percolation thresholds or other critical points or even critical exponents from finite samples is a pervasive problem in statistical physics, and involves consideration of the correlation length and its relation to sample size, as discussed in detail for sea ice X-ray tomography in (Pringle et al., 2009). One of my concerns about this paper is that there does not appear to be any consideration at all of the relationship between the one sample size they look at and their conclusions. Perhaps samples with vertical extent of 8 cm (if they could have scanned those) might typically require a brine volume fraction of 3.5% for there to be connections from top to bottom which include the micro-scale features that have been resolved with their instrument and analysis. Figure 3 in [Pringle et al., 2009] shows a transition around brine volume fraction of about 5% in the behaviour of the fractional connectivity (fraction of brine voxels on one face connected to the opposite face, regardless of path characteristics) for sea ice single crystals, and its dependence on sample size, and a corresponding divergence of the correlation length as 5from below. Do you have data below 2.4% that shows a similar transition or correlation length divergence as you approach the threshold from below, which would.*

**Answer:**
Our conclusions are based on two samples sizes - the centrifuge experiments are done with core sub-samples of 7.25 cm diameter and 3.5-4 cm thickness. This sample size is more or less a standard in sea ice property analysis. Based on these samples we obtain a critical exponent $\beta$ between connected/centrifuged and total porosity that is very close to the theoretical prediction by directional percolation theory, and a percolation threshold. We then show that the results from CT-imaged sample sizes (2 cm horizontal and 0.55 cm vertical) are more scattered but fully consistent. With regard to larger sample sizes it can be first noted that the permeability results from Freitag (1999) were obtained for samples sizes of 6 cm vertical dimension and 9 cm diameter. These agree with our results, however they do not extend to low porosities. **We have now added one more figure (Fig. 13)** to discuss possible sample scale effects based on our results. Comparing the permeability and its threshold behaviour for 0.55 cm with harmonic means for the 2-3 cm vertical sample length does not show a significant difference. For larger vertical extents the proportion of inclined channels that leave the sample laterally will affect the results.

We pointed out we suspected that, considering the 0.07 mm threshold we found, the results from Pringle et al. (2009) may have been limited by their CT image resolution (with a Nyquist criterion of 0.083mm). We agree that, with regard to the correlation length problem, data below the 2.4% threshold could be useful, which we do not have. We also agree that smaller scales could be worth investigating with higher resolution, especially as our spatial resolution (Nyquist criterion 0.036 mm) was just a factor of two

smaller than the 0.07 mm threshold. However, as outlined in the discussion, the necking transition near 0.07 mm is consistent with what others have observed, which indicates that it indeed could be the critical length scale for the permeability of young ice.

For easier reading we divide it into several sub-comments with answers.

*2a. For binary lattice percolation models, such as the 2D square lattice with bonds open or closed, given a finite sample and a bond configuration, there either is or is not a path of open bonds connecting one side of the sample to the other. However, if the bonds are pipes with arbitrarily small radii, that is, the radii are chosen from a probability distribution with support down to 0, then the question of whether a configuration or cluster percolates or not is now determined by how 'thick' one requires the spanning pathways in the cluster to be. In other words, given a cut-off radius, one can then ask if connected clusters of pipes whose radii exceed the cut-off percolate or not. If they do, then fluid flowing through a percolating cluster of large enough pipes will generally be forced to travel through some of the smallest pipes whose radii are near or at the cut-off. Moreover, these 'bottlenecks' or throats determine the leading order behaviour of the fluid permeability, or the effective electrical conductivity if the bonds are conductors. There are rigorous theorems (and analogous techniques in theoretical solid state physics) to this effect that form the basis of critical path analysis [Golden and Kozlov, in Homogenization: Serguei Kozlov Memorial Volume, V. Berdichevsky et al. (Eds.), 1999; Golden, in Homogenization and Porous Media, U. Hornung (Ed.), 1997; Golden et al., GRL, 2007; Ambegaokar, Halperin and Langer, Phys. Rev. B, 1971]. In the context of sea ice, which is of course a continuum material, the percolation characteristics of the porous brine microstructure can be thought of in terms of the pipe network described above. One way of putting a principal result of this paper is that in sea ice samples of vertical dimension 2 cm to 3 cm with brine volume fractions exceeding 2.4%, there are fluid pathways through the brine phase connecting the top to the bottom whose minimal 'diameter' exceeds 0.07 mm (or in terms of the pipe network, configurations of pipes whose diameters exceed a cut-off of 0.07 mm span the sample vertically, or percolate).*

**Answer:**
We think that we intended to have pointed it out as one of our main results (see abstract, the discussion subsection 4.3 and the conclusions): we find that our permeability threshold is related to necking of pores near a diameter of 0.07 mm. We think that this may be a basic property of sea ice microstructure and phase transitions that can be derived from our young ice study.

*2b. Now, let us discuss how this result is related to the so-called 'rule of fives' and the generally accepted value of 5% brine volume fraction for the 'percolation threshold' of sea ice. The concise statement of this 'rule of thumb' in the first paragraphs of the papers [Golden et al., Science, 1998; Golden et al., GRL, 2007, Pringle et al., JGR, 2009] is that columnar sea ice is 'effectively impermeable' to bulk fluid flow for brine volume fractions below about 5%. This is not stated as a mathematical theorem, and there is an understanding that by the very nature of percolation theory for finite samples, and the complex multiscale structure of the brine phase, one would expect the possibility of some fluid flow over relatively small scales through relatively small pore spaces, even for brine volume fractions below 5%. (Figure 3b in [Golden et al., 2007] shows that the fractional connectivity for samples of 8 mm in vertical extent remains non-zero down to below 4%*

*brine volume fraction.)*

**Answer:**
The abstract from Golden et al. (1998) states 'For temperatures warmer than Tc, brine carrying heat and nutrients can move through the ice, whereas for colder temperatures the ice is impermeable'. It does not state 'effectively impermeable'. It further claims that 'The similarity of sea ice microstructure to compressed powders is used to theoretically predict $p_c$ (the threshold) of about 5 percent'. At that time a better estimate than 'about 5 percent' was not possible, simply because data and model approaches were not detailed enough. Also, as noted in our discussion, Golden et al. (1998) used the permeability data from Ono and Kasai (1985) to support the 5% threshold and 'rule of the fives'. This ice, however, was very young and thus very likely had much higher salinity and porosity then first-year ice. We think that it is in time to revise the conclusions drawn in the early account of the problem by Golden et al. (1998), and to provide results based on detailed data and modelling as we have done here.

*2c. However, I have personally made hundreds of in situ measurements of the vertical fluid permeability of sea ice in the Arctic and Antarctic, by removing partial cores and then measuring the rate at which water fills the hole through the ice at the bottom of the hole by various techniques. Even with the uncertainties in this 'sack hole' method, I can unequivocally state that if the sea ice at the bottom of the hole is columnar and has brine volume fraction below about 4% or 5% (and horizontal flow is blocked with 'packers'), there will most likely be very little or no measurable fluid in the hole even after a few hours. As the brine volume fraction of columnar sea ice decreases from high values associated with quite permeable ice, there is a noticeable, clear transition to bulk fluid flow over the scale of tens of centimetres relevant to the experiment, being essentially shut down, or the ice becoming effectively impermeable, once the brine volume fraction gets below about 5%. Roughly speaking, permeability values then generally lie below about $10^{-12}$.*

**Answer:**
As pointed out in the last answers, and in the manuscript, a closer look into the data on which the 'rule of the fives' once was based indicates inconsistencies: Permeability data of Ono and Kasai (1985) relate likely to much higher porosities than once assumed, while CT-based results from Pringle et al. (2009) could be resolution limited. Our analysis provides new insights for young ice. However, there is need for measurements of permeability and determination of porosity thresholds for older ice. However, to our knowledge most of the 'hundreds of in situ measurements of the vertical fluid permeability' the reviewer mentions have not been published. Those that we are aware of are published in Fig. 1 of Golden et al. (2007) and show considerable scatter. Putting an 'effective permeability threshold' near a porosity of 4 or 5% and saying that 'permeability values then generally lie below about $10^{-12} m^2$' seems not well backed up by data. Have we overlooked a publication?

*2d. The spirit in which this rule was developed was in terms of whether or not the brine microstructure could enable various geophysical processes such as surface flooding and subsequent snow-ice formation, melt pond drainage, and changes in salinity. For example, suppose we consider upward percolation of sea water and brine due to snow loading of the ice surface, and the subsequent freezing of the flooded surface snow. If the upper layer of sea ice through which fluid must pass to reach the surface, say, has*

*permeability around $10^{-13}$ with brine volume fraction just below the threshold, we may wind up with very little water on the surface and essentially no new snow-ice production. On the other hand, if the permeability of this restrictive layer is around $10^{-11}$ or larger with brine volume fraction a bit above the threshold, then after several hours there may be a few centimetres of water on the surface which could produce a significant amount of snow-ice that affects ice mass-balance accounting.*

**Answer:**

We think that, even if movement is slow at low porosities, a more precise porosity threshold and permeability prediction would be valuable for many sea ice problems: E.g., for modelling slow desalination, gas and nutrient fluxes on time scales of months and years. This contrasts more rapid flow processes like surface flooding and melt pond drainage. Moreover, this slow fluid flow will change the microstructure on which it depends, and thus can be important for the evolution of many physical properties like albedo, strength and thermal conductivity and also: the permeability itself. Hence, there is good reason to revise the threshold and obtain more accurate estimates that are linked to the microstructure. To make progress, percolation theory is a powerful tool, but its validation for sea ice needs to be based on detailed microstructure observations. There are many processes in need of the intrinsic 'permeability' and not only the 'effective permeability'.

*2e. From the point of view of the pipe network, the 5% threshold for bulk flow in practice means that in sea ice samples of vertical dimension on the order of, say, 20 cm to 50 cm with brine volume fractions above about 5%, there are fluid pathways through the brine phase connecting the top to the bottom whose minimal 'diameter' exceeds a certain cut-off value, which is much larger than 0.07 mm. As reported in [Weeks and Ackley, 1982], S. Martin and co-workers over many studies found that most vertically oriented 'channels' through which the bulk of fluid is transported through sea ice over tens of centimetres have diameters that range from about 1 mm to 1 cm, with some channels much larger. In Equation (4) of [Golden et al., GRL, 2007] the cut-off, bottleneck, or minimal diameter was then chosen to be 1 mm, which leads via critical path analysis to the prediction of the scaling factor in front. This percolation formula in Equation (4) with a bulk transport threshold of 5% then agrees very closely with in situ data for brine volume fractions above the threshold. By way of comparison with the scales considered in the current paper, the amount of fluid that flows per unit time through a circular pipe of diameter 1 mm is $10^4$ times the amount that flows per unit time through a pipe of diameter 0.1 mm, which is just a bit larger than the critical diameter of 0.07 mm in this paper.*

**Answer:**

These 'pipe networks' are secondary channels that evolve over time in sea ice. Without doubt, in older and warmer sea ice their diameters are much larger than 0.1 mm. Our study has been concerned with young ice, where such wider brine channels systems had only weekly developed. To our knowledge, predicting their evolution is a challenging and unsolved problem. However, whatever their diameter is, they do not evolve on their own, but are fed from and formed out of the smaller scale channel networks. This makes the microstructure and permeability of young ice so important: it is the starting point for desalination and formation of wider brine channels networks.

*2f. Thus, it is not really appropriate to state without careful explanation and qualification that the 2.4% value found here is considerably lower than the 5% threshold which has been widely used for bulk flow over larger scales. What is being referred to is quite different for these two situations, in the setting of a multiscale porous medium like sea ice as described above, with the rule of fives and the 5% threshold describing fluid transport behaviour on significantly larger sample and pore size scales than what is considered in this paper. (In fact, the vertical sample size of 5 mm for the permeability simulations calls into question the applicability of this work beyond the smallest of scales. See Figure 3 in [Pringle et al., 2009] on the dependence of the correlation length with brine volume fraction.) The results described in [Golden et al. 2007, Pringle et al., 2009] with a vertical threshold of 4.6% (and higher thresholds in the horizontal directions) were a first step in imaging the connectivity of the brine phase and building toward larger scales, with the analysis of the most basic building blocks - sea ice single crystals. As stated in [Golden et al. 2007], 'These images provide insight into and constraints for more detailed modelling of micro-scale inclusion connectivity. A similar analysis of large-scale pore networks remains challenging, but is inherently reflected in the in situ permeability data.' Indeed, extending such analyses to the scales relevant for the rule of fives remains a challenge today.*

**Answer:**

The point that we make and like to stress again is that observations and mathematical models so far have been incomplete to properly constrain or support a 5% threshold, in particular for thin young ice, or the near bottom regime of thicker ice. And it is for such young ice for which the 'rule of the fives' has once also been proposed Golden et al. (1998); Pringle et al. (2009). We agree that the analysis of older sea ice, containing wider brine channel networks, is a challenge. Yet we note that in our opinion, looking critically at the available data for older ice, we cannot see a clear evidence that the 'rule of the fives' should be valid there. From the viewpoint of pore microstructure, there is too little information about volume fractions of the pore space that belong to larger and smaller pores, and how they are connected.

To account for the 2a-2f comments and answers in the manuscript, we have rewritten subsection 4.5 on the porosity threshold and added a section 4.6., where we clarify the need to distinguish between granular and columnar ice texture, young and old ice and local and full-thickness permeability, and discuss scale effects. We point out that our results only apply to columnar young sea ice, and add figure 10 and its description in 3.4 to make the overall pore size distribution of our ice more clear. We also modified the abstract, see above.

[revised manuscript text omitted]

*3. One major significance of the paper, in my opinion, is that they have explored a new level of fine scale structure and conducted a high resolution analysis of the habitats of microbial life in sea ice, and also carefully computed a key property of sea ice which is critical for local nutrient fluxes, namely, fluid permeability, on scales which may be particularly relevant for small scale biological processes.*

**Answer:**

We agree, but like to add that the fine scale results established here are also essential for sea ice desalination and microstructure evolution in general, being a starting point to ultimately predict physical properties and their evolution. To make this clearer, and indicate future needs, we have added the following paragraph in the conclusion:

L574 The present work presents new insight into the sea ice pore space evolution of young sea ice and theoretical interpretation of the latter. It demonstrates the large potential of 3D X-ray micro-tomographic imaging to make progress in our fundamental understanding of sea ice properties. For future work we suggest several directions to make further progress. First of all, the result for permeability-porosity relations and thresholds, are not directly transferable to the thicker, older and warmer summer ice. The latter often contains coarser secondary brine channels that are lacking in young ice and that are relevant for processes melt pond percolation and melt pond albedo feedbacks (Freitag and Eicken, 2003; Polashenski et al., 2017). This reflects one of the challenges in sea ice physics, which is to improve our understanding how the sea ice pore space, as well as permeability and other physical properties evolve over time. To make progress more 3D CT data of sea ice and its pore space evolution over time are needed. Second, while the present study may be seen as a starting point to a concise understanding and modelling of this evolution, it should be verified with higher spatial resolution to clarify any resolution limit with respect to necking and porosity thresholds. Third, there is a need for comparing granular and columnar ice, as the granular surface layer will be important for the through-flow permeability. And last but not least, due to the lack in experimental data, carefully controlled laboratory experiments in the lines of Ono and Kasai (1985) would be of high value. Combing such experiments with repeated CT imaging to monitor flow-induced microstructure changes, could provide valuable insight about the evolution of the sea ice pore space and its permeability. A useful concept with centrifuged

sea ice would be measurements of kerosene permeability in a permeameter (Saito and Ono, 1978; Saeki et al., 1986), allowing validation of permeability computed from CT imagery and the question if there are sub-resolution pathways.

*4. I am concerned that while centrifuging may leave major inclusion structures intact, if this process modifies the brine microstructure by opening up new pathways that then appear to be connections in the X-ray tomographic images, it will be at these finest scales that are the focus of this paper. I think there should be some discussion of how the centrifuging process may or may not affect the 2.4%.*

**Answer:**
Freitag (1999) has performed the same centrifuge technique, and has discussed some aspects. We add now the following paragraphs sentence in the methods and the discussion: Methods, L89 The centrifuge parameters depend on centrifuge type and were carefully chosen on the basis of several tests. (i) Ice samples were centrifuged with temperature loggers to determine temperature stability. Slight warming of the centrifuge was observed, leading us to the choice of an in initial centrifuge temperature 1K below the ice in-situ temperature. A similar value was chosen by Weissenberger et al. (1992) similar centrifuge times. (ii) Varying the centrifuging time from 10 to 20 minutes showed that more than 95% of brine where extracted during the first ten minutes, and we selected 15 minutes. (iii) Freitag (1999) noted that incomplete centrifugation of brine might lead to brine remnants which, after cooling and freezing, might block pores and decrease the permeability. We have indeed found such a result in an earlier study with centrifuge acceleration of 15 g (Buettner, 2011) and thus tested the effect of relative centrifuge acceleration for three ice cores at 10, 25 and 40 g. The result was on average 20% less centrifuged brine at 10 g, but only a slight non-significant 5% difference between 25 and 40 g. We thus are confident that 40 g is a proper choice for extracting the connected brine.

Discussion, L368 We have considered and avoided several possibilities how centrifugation might bias the results. Incomplete centrifugation of brine might lead to brine remnants which, after cooling and freezing, might block pores (Freitag, 1999). This might create a higher apparent porosity threshold, indicated by an earlier study Buettner (2011) with lower centrifuge acceleration (15 g compared to our 40 g). By carefully choosing the parameters we think that we largely avoided this problem. Also the warming of ice samples in the centrifuge was carefully tested and avoided by using a centrifuge start temperature 1 K below the in-situ sea ice value. Other effects, like pressure melting of ice or internal deformation, are unlikely at the relatively small centrifuge rates we used. We cannot exclude that centrifuging has implied minor deviations from in situ temperatures. However, what we derive, in essence and for the first time, from centrifuging and CT imaging, is the relationship between open porosity, total porosity and permeability. We rate it as unlikely that small internal structure changes due to fluid redistribution and freezing/melting in the centrifuge will change this relationship fundamentally.

*5.The results in [Perovich and Gow, JGR, 1996] should be referenced and briefly discussed in light of the results in this paper.*

**Answer:**

We add in the discussion:

L428 **Also Perovich and Gow (1996) have optically analysed sea ice inclusions in 2D thin sections, focusing however on other microstructure characteristics. From their tabulated values of major axis length, perimeter and circularity of ellipses that were fitted to brine pores, one can deduce a minimum axis length. Median values obtained in this way (see Maus (2007)) fall in the range of 0.05 to 0.1 mm and are comparable to our observed values. However, from the 2D data no information on pore connectivity and necking is available.**

*6. A few sentences, or synopsis, about the scales of features the authors can resolve and how their resolution compares to previous works would be welcome, or if this were made a little clearer with a sentence like: 'The inclusions we see that form connected fluid pathways at lower brine volume fractions than have been observed before have the following characteristic ranges of dimensions, etc.' The paragraph around line 135 needs to be expanded and clarified.*

**Answer:**

We added:

L136 **With the current imaging settings, and image processing for analysis and simulations, we expect to observe pores and inclusions with smallest dimensions of 36 $\mu$m (corresponding to a Nyquist criterion of 2 times the voxel size of 18 $\mu$m). This is an improvement by a factor of two compared to the voxel size of $41.5\mu$m in the CT-image study of laboratory grown ice by Pringle et al. (2009). Our horizontal scale is large enough to also observe pores and brine channels situated between grains of typical dimensions 5 to 20 mm. Our standard vertical scale of 5.5 mm is smaller than used in standard sea ice bulk sample analysis of several centimetres, yet we can always merge the subsamples to look at comparable vertical scales. The choosen horizontal and vertical scales are well above the typical pore scale characteristics of young ice obtained by Eicken et al. (2000) based on the analysis of Magnetic Resonance Images with lower resolution (0.09mm voxel size).**

*7. In line 375, it is stated, 'So far sea ice permeability has been studied in terms of isotropic percolation (Petrich et al., 2006; Golden et al., 2007; Pringle et al., 2009)'. However, in [Pringle et al., JGR, 2009] the finding of different values of the percolation threshold in three perpendicular directions for a sea ice single crystal is certainly anisotropic percolation, and should be noted as such.*

**Answer:**

In the context of this statement we had noted that Pringle et al. (2009) have analysed their data based on isotropic percolation theory, while directed percolation belongs to a different universality class (L380-381). However, to make this difference clearer we add at the end of the paragraph:

L383 **E.g., it will be a future challenge to study the anisotropy in permeability, observed by Freitag (1999), and the directional dependence of the porosity**

threshold, found by Pringle et al. (2009), in terms of *directed percolation.*

*8. The idea of analysing the effective centrifuged brine volume fraction compared to the total is excellent. However, by a certain point there seem to be so many different types of porosities running around that it is difficult to keep them straight. Perhaps a synopsis and overall explanation would be helpful, as well as clearer definitions with diagrams of all the parameters in Table 2.*

**Answer:**
We considered this but decided that the information in Table 1 should be sufficient. In essence we only focus on the dependence of all properties on total porosity.

*9. Line 23 - should refer to [Polashenski et al., JGR, 2018] which deals explicitly with the issue of how initially permeable sea ice supports melt ponds.*

**Answer:**
We added this reference in the extended conclusion (see our response to 3. above, new paragraph at L574)

*10.Line 92 - missing a 'c' in subscript of phi.*

**Answer:**
Corrected

*11. The exponent in Equation (8) is 2.6, which I assume is a best fit. The corresponding exponent in Equation (4) in [Golden et al., GRL, 2007] is 2. This is a theoretical prediction, based on an argument that even though sea ice is a continuum material that could exhibit so-called non-universal behaviour different from lattices (with its exponent larger than 2, see [Golden, PRL, 1990] for rigorous results on lattices), it exhibited universal behaviour due to the log-normal distribution of brine inclusion sizes. It is interesting to speculate if the exponent of 2.6 in this paper is a demonstration of non-universal behaviour at these fine scales. Perhaps a sentence could be added addressing this?*

**Answer:**
We felt the need to discuss the exponent in some detail. E.g., we think that a relationship between pore sizes and porosity will basically create a larger exponent than 2. We added a discussion of the permeability exponent in section 4.4:

L512: **For modelling efforts of the permeability with porosity, the obtained exponents are of high relevance. In general the dependence of permeability $K$ on porosity $\phi$ is often empirically characterised by an equation of the form $K \sim \phi^b$. The simplest model is to relate porosity $\phi$ to pore diameter $d$ and assume the well known relationship $K \sim \phi d^2$. From the basic sea ice microstructure of parallel brine layers ($d \sim \phi$) and circular tubes ($d \sim \phi^{1/2}$) one could argue for $2 < b < 3$. The larger exponent $b = 4.0 \pm 0.4$ that we find (Figure 7) reflects that not only pore diameters are changing with porosity but also their connectivity. Percolation theory accounts for this effect, which is resembled by an equation that involves the threshold porosity in the form $K \sim (\phi - \phi_c)^t$ (see also Eq. 8). For isotropic percolation this has been first proposed as 'critical path analysis' for the electrical conductivity Ambegaokar et al. (1971), and the current best estimate for the conductivity exponent, in the form $E \sim (\phi - \phi_c)^e$**

is $e = 2.0$ (Stauffer and Aharony, 1992). The permeability exponent $t$ however will be larger than $e = 2.0$ if the characteristic pore scale $d_c$ of permeating paths is changing with porosity. This problem has been addressed by several authors proposing an equation of the form $K \sim d_c^2 (\phi - \phi_c)^e$. E.g., Katz and Thompson (1986) proposed to determine the length scale $d_c$ on the basis of mercury porosimetry. This is indeed the approach (virtual porosimetry) we have applied to determine the throat sizes in the present study. If we insert the throat size exponent from Fig. 10b, $d_c \sim \phi^{0.46}$, and assume $e = 2.0$ we would get a dependence of the form $K \sim \phi^{0.92} (\phi - \phi_c)^2$. This in turn is not very different from our percolation-based fit $K \sim (\phi - \phi_c)^{2.6}$. We note that Golden et al. (2007) have proposed $t = 2.0$ for older sea ice, arguing that wide brine channels with a constant critical diameter control the permeability, and also reported $t = 1.97$ as a best fit to the data from Ono and Kasai (1985). We were unable to reproduce the latter result based on the data in Figure 11 (new 12), using our estimates of ice salinities in the experiments from Ono and Kasai (1985). Our best fit for the permeability critical exponent, $t = 2.6 \pm 0.3$, is largely consistent with percolation theory, critical path restriction by throats and their dependence on porosity. It is valid for young sea ice where the permeating pores are shrinking with decreasing porosity, and will differ for ice with a different pore-porosity relationship. Also for young ice, a general prediction may be more complicated due to three aspects: First, it has been pointed out by Le Doussal (1989) that the approach from Katz and Thompson (1986) needs to be revised for broad pore size distribution which may lead to higher exponents for the permeability. Second, the exponents may also be different for directed percolation. And third, ice type may play a role and results for granular ice may be different. We are currently investigating this problem in more detail.

12. *Line 15 - I believe the authors meant to say 'Sea ice is a porous medium that covers, on average, about 12 percent of the earth's oceans.' (Or about 7 percent of earth's surface).*

**Answer:**
'5-7 percent of the earth's oceans' is indeed what we ant to say. There is, in the present climate, not more sea ice covering the oceans at the same time, considering that a sea ice maximum in the northern hemisphere coincides with a minimum in the southern hemisphere and vice versa.

[Figure]

Figure 1: **New Figure 10** Pore size distributions based on XRT imaging of 4 young ice cores. a) Fraction of open brine pores in 18 $\mu$m wide size bins for the two warmest (red) and two coldest (blue) cores. The corresponding cumulative fractions are also shown, with y-axis on the right hand side; b) same as a) but for the porosimetry/ fraction of pores throats.

[Figure]

Figure 2: **New Figure 13** Comparison of simulated vertical permeability $K$ and $(\phi - \phi_c)$ for columnar samples (as shown in Figure 11 and here as open circles) to two other simulation results. The solid circles are harmonic means of all subsamples (normally four to five) from each basic sample cut in the field (and centrifuged). The red stars are permeability results for near-surface samples (up to 5 cm from the surface) that were classified as granular ice and thus excluded from the basic analysis.